# Towards complete assignment of the infrared spectrum of the protonated water cluster H$^+$(H$_2$O)$_{21}$

Jinfeng Liu [1,2,8], Jinrong Yang [2,8], Xiao Cheng Zeng [3], Sotiris S. Xantheas [4,5✉], Kiyoshi Yagi [6✉] & Xiao He [2,7✉]

The spectroscopic features of protonated water species in dilute acid solutions have been long sought after for understanding the microscopic behavior of the proton in water with gas-phase water clusters H$^+$(H$_2$O)$_n$ extensively studied as bottom-up model systems. We present a new protocol for the calculation of the infrared (IR) spectra of complex systems, which combines the fragment-based Coupled Cluster method and anharmonic vibrational quasi-degenerate perturbation theory, and demonstrate its accuracy towards the complete and accurate assignment of the IR spectrum of the H$^+$(H$_2$O)$_{21}$ cluster. The site-specific IR spectral signatures reveal two distinct structures for the internal and surface four-coordinated water molecules, which are ice-like and liquid-like, respectively. The effect of inter-molecular interaction between water molecules is addressed, and the vibrational resonance is found between the O-H stretching fundamental and the bending overtone of the nearest neighboring water molecule. The revelation of the spectral signature of the excess proton offers deeper insight into the nature of charge accommodation in the extended hydrogen-bonding network underpinning this aqueous cluster.

[1] Department of Basic Medicine and Clinical Pharmacy, China Pharmaceutical University, Nanjing 210009, China. [2] Shanghai Engineering Research Center of Molecular Therapeutics and New Drug Development, School of Chemistry and Molecular Engineering, East China Normal University, Shanghai 200062, China. [3] Department of Chemistry, University of Nebraska, Lincoln, NE 68588, USA. [4] Advanced Computing, Mathematics and Data Division, Pacific Northwest National Laboratory, 902 Battelle Boulevard, P.O. Box 999, MS K1-83, Richland, WA 99352, USA. [5] Department of Chemistry, University of Washington, Seattle, WA 98195, USA. [6] Theoretical Molecular Science Laboratory, Cluster for Pioneering Research, RIKEN, 2-1 Hirosawa, Wako, Saitama 351-0198, Japan. [7] New York University-East China Normal University Center for Computational Chemistry, New York University Shanghai, Shanghai 200062, China. [8] These authors contributed equally: Jinfeng Liu, Jinrong Yang. ✉email: Sotiris.Xantheas@pnnl.gov; kiyoshi.yagi@riken.jp; xiaohe@phy.ecnu.edu.cn

The microscopic nature of an excess proton interpenetrated within a three-dimensional hydrogen bonded network of water remains a long-standing elusive aspect of aqueous acids, mainly due to the inherent spectral complexity of bulk water[1–7]. Gas-phase aqueous clusters with an excess proton of precisely controlled compositions, $H^+(H_2O)_n$, thus offer useful bottom-up model systems that enable one to focus on the evolution of vibrational spectral features associated with the excess proton surrounded by a well-defined number of water molecules[1,8–15]. In this study we introduce a new protocol based on the combination of high level electronic structure theory and the inclusion of anharmonicity, and demonstrate that it is able to obtain the nearly complete assignment of the infrared (IR) spectrum of the magic number $H^+(H_2O)_{21}$ cluster in excellent agreement with experiment.

Although the properties of isolated clusters are much simpler than those of the bulk liquid, their spectroscopic features are still complex, and the unequivocal interpretation of the spectroscopic signature of an excess proton in water clusters hinges on synergetic experimental and theoretical works[16–20]. Theoretical calculations of the IR spectra on candidate local-minimum structures aid the assignment of the vibrational features with respect to the experimental observations. The small-sized protonated water clusters, containing no more than 11 water molecules, have been extensively studied through experimental and theoretical works in the past decades[16,17,19–23]. The proton in water cluster has been conventionally considered to be in two accommodation motifs, the Eigen form[21] (i.e., a hydrated hydronium cation, $H_3O^+(H_2O)_3$) and the Zundel form[16] (i.e., a proton shared between two water molecules, $H_2O\cdots H^+\cdots OH_2$), which would induce dramatically different vibrational features near 1000 and 2660 cm$^{-1}$, respectively, in the IR spectrum[12]. Agmon and co-workers have studied the IR spectra of protonated water clusters of different sizes by using ab initio molecular dynamics simulations[24–27], which contributed comprehensive understanding of the protonated water structures.

New spectral signatures emerge as the small size cluster grows into a complex three-dimensional network in cage morphology with the increasing number of water molecules[13]. A long-standing puzzle regarding the evolution of the vibrational features in the intermediate size regime is the emergence of a pronounced intensity anomaly at the "magic number" size, $H^+(H_2O)_{21}$[1,15,28–31]. Many previous studies have suggested that the $H^+(H_2O)_{21}$ cluster takes shape of a configuration in which an Eigen-state cation $H_3O^+$, whose O-H bonds are hydrogen-bonded to three $H_2O$ molecules, tends to reside on the surface of a dodecahedral cage containing one interior $H_2O$ molecule[29,30,32–35], as shown in Fig. 1. However, the link between the experimental IR spectrum and the theoretically predicted structure was missing since the computed IR spectra did not match the experiment. The harmonic normal-mode analysis based on density functional theory (DFT) predicted strong IR bands of the symmetric and asymmetric O-H stretching modes of the cage-surface-bound $H_3O^+$ near 2600 cm$^{-1}$, which were not evident at all in the experimental spectrum[1]. Later, Torrent-Sucarrat and Anglada[30] have shown that the anharmonic coupling plays a crucial role in the characterization of the IR spectrum of the $H^+(H_2O)_{21}$ cluster. Their calculation by the second-order vibrational perturbation theory (VPT2) predicted a strong asymmetric O-H stretching band of the $H_3O^+$ around 2000 cm$^{-1}$, outside the region measured in the experiment. Recently, Fournier et al.[10] have successfully extended the range of the measurement down to 600 cm$^{-1}$, and found that the agreement between the experiment and theory was only qualitative. One of the reasons for the discrepancy was the low theoretical level treatment of the anharmonicity in predicting the IR spectrum of the $H^+(H_2O)_{21}$ cluster. Yu

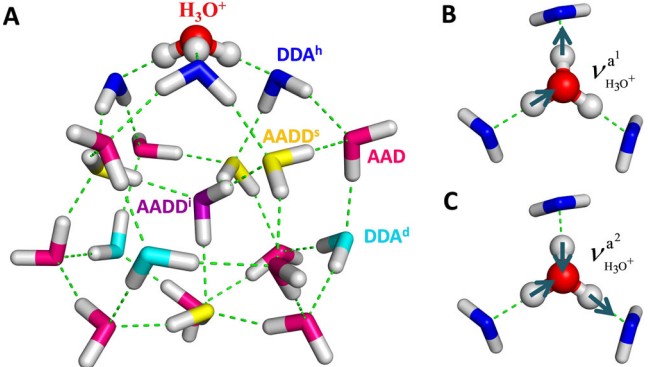

**Fig. 1 Representation of the $H^+(H_2O)_{21}$ cluster. A** The optimized Eigenstate $H^+(H_2O)_{21}$ structure using the fragment-based CCD/aug-cc-pVDZ level of theory. $H_3O^+$ is at the top of the structure. There are two types of DDA water molecules: three are hydrogen-bonded with $H_3O^+$ (blue, DDA$^h$) and three are distant from $H_3O^+$ (turquoise, DDA$^d$). Nine AAD-type water molecules are colored in pink. Four-coordinated AADD water molecules are colored in purple and yellow, respectively, corresponding to the one in the interior (AADD$^i$) and four at the surface (AADD$^s$). **B, C** represent two asymmetric O-H stretching modes of $H_3O^+$, denoted $\nu^{a1}_{H_3O^+}$ and $\nu^{a2}_{H_3O^+}$.

and Bowman[36] have shown that the higher level, vibrational configuration interaction (VCI) method achieves a significant improvement over VPT2. Nevertheless, the calculation was based on a potential energy surface (PES) of $H^+(H_2O)_{21}$ represented as a sum of many-body potential energy functions (PEFs) of small clusters, $H^+(H_2O)_n$ ($n = 1–4$)[37–39] and $(H_2O)_n$ ($n = 1–3$)[40], derived from ab initio electronic structure calculations. VCI calculations were also carried out for fragments of the cluster, $H_3O^+(H_2O)_3$ (15 dimensions) and each $H_2O$ (3 dimensions). Theoretical calculations that account for both the electronic and vibrational structures of the full $H^+(H_2O)_{21}$ cluster remain a challenge.

Herein, we report a new protocol for computing the IR spectra of complex molecular clusters that has the potential of establishing a transformative opportunity in the field. We apply this protocol to the "guinea-pig" of the field, namely the IR vibrational spectrum of the $H^+(H_2O)_{21}$ cluster with an Eigen structure. We rely on the state-of-the-art, fragment-based Coupled Cluster (CC) theory[41,42] and the Second-order Vibrational Quasi-Degenerate Perturbation Theory (VQDPT2)[43,44]. Our previously developed electrostatically embedded generalized molecular fractionation (EE-GMF) method[41], whose accuracy and efficiency have been rigorously evaluated in a series of studies, is utilized to reduce the computational scaling of the full system CC calculations. EE-GMF shows an acceleration by factors of >40 over the conventional full system calculations, while the deviations of EE-GMF calculated energies of systems containing over 100 water molecules at diverse ab initio levels are mostly within 0.01 a.u. as compared to the full system calculations[41,45]. VQDPT2 has been tested to be as accurate as VCI for small molecules[43,44], but is scalable to many-mode systems. Recently, the method has been further improved by utilizing local coordinates and applied to strongly hydrogen bonded network in biomolecules[46]. In this work, VQDPT2 has been carried out in 89 dimensions using coordinates localized to each molecule of the $H^+(H_2O)_{21}$ cluster (See the Methods and Supplementary Methods sections for details). The combination of high-level quantum electronic and vibrational calculation yields accurate spectral features compared to experiment, thus resolving the physical picture of an excess proton accommodation in this complex water network.

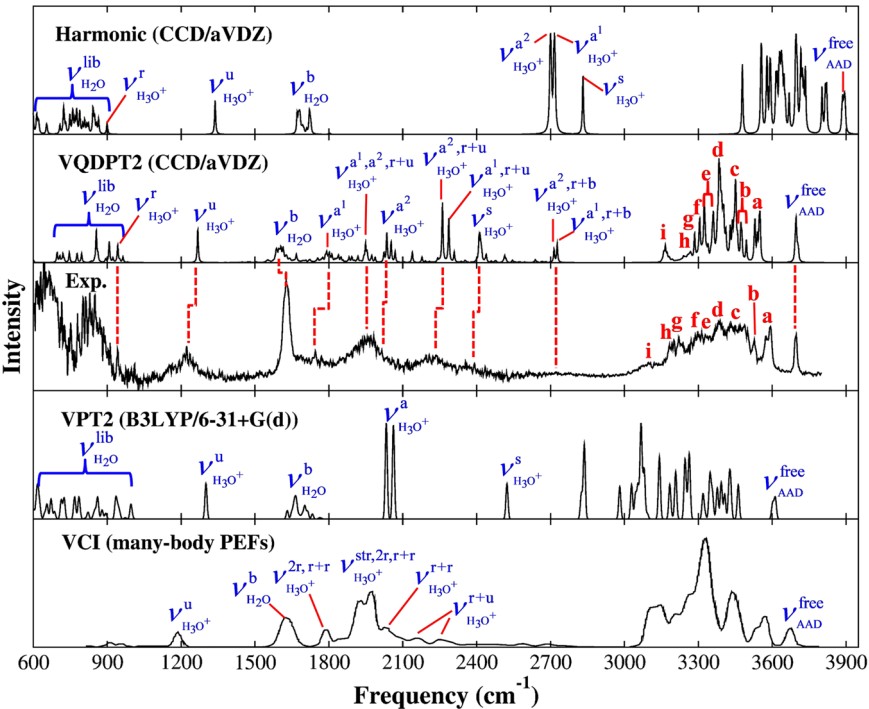

**Fig. 2 IR spectra of the H⁺(H₂O)₂₁ cluster.** IR spectra of the $H^+(H_2O)_{21}$ cluster obtained by the harmonic approximation and VQDPT2 method based on the fragment-based CCD/aug-cc-pVDZ approach, as compared to the experiment[10] and the previous calculations by VPT2 based on B3LYP/6-31+G(d)[30] and VCI based on the many-body PEFs[36]. The fundamental excitations of $H_3O^+$ are denoted $\nu^r_{H_3O^+}$ (frustrated rotation), $\nu^u_{H_3O^+}$ (umbrella vibration), $\nu^{a1}_{H_3O^+}$ and $\nu^{a2}_{H_3O^+}$ (asymmetric O-H stretching), $\nu^s_{H_3O^+}$ (symmetric O-H stretching), and those of $H_2O$ are denoted $\nu^{lib}_{H_2O}$ (libration), $\nu^b_{H_2O}$ (bending), $\nu^{free}_{AAD}$ (dangling O-H stretching of AAD), and (**a-i**) (O-H stretching). The resonance states between the fundamental excitation of asymmetric O-H stretching and the combination tones (r+u and r+b) are denoted $\nu^{a1,a2,r+u}_{H_3O^+}$, $\nu^{a1,r+u}_{H_3O^+}$, $\nu^{a2,r+u}_{H_3O^+}$, $\nu^{a1,r+b}_{H_3O^+}$, and $\nu^{a2,r+b}_{H_3O^+}$, where r, u, and b represent the frustrated rotation, the umbrella vibration, and the HOH bending of $H_3O^+$, respectively. The harmonic and VQDPT2 spectra are broadened using Lorentz functions with the full-width at half-maximum (FWHM) of 5 cm⁻¹. The raw stick spectrum is shown in Supplementary Fig. 1.

## Results

**The structure of H⁺(H₂O)₂₁.** The optimized $H^+(H_2O)_{21}$ structure at the fragment-based Coupled-Cluster Doubles level with the aug-cc-pVDZ basis set (CCD/aug-cc-pVDZ) is color-coded in Fig. 1A for a better clarification of the assignments of the vibrational bands. The Eigen-type hydronium ion (red sphere), integrating three hydrogen-bonded DDA water molecules (blue, denoted DDAʰ, where A and D stand for hydrogen-bond acceptor and donor, respectively), is located on the surface of the cage. Each of the DDAʰ water molecules donates a hydrogen bond to an AADD (yellow) and an AAD (pink) water molecule, respectively. The upper three AAD water molecules also connect to three lower DDA water molecules, which are all distant from the $H_3O^+$ cation (turquoise, denoted DDAᵈ). There are five four-coordinated AADD water molecules, four of them on the surface (denoted AADDˢ) and the remaining one in the interior (denoted AADDⁱ).

**Theoretical and experimental IR spectra.** The harmonic and VQDPT2 spectra computed at the fragment-based EE-GMF CCD/aug-cc-pVDZ level are shown in Fig. 2, along with the previous theoretical spectra obtained by the VPT2 method based on DFT at the B3LYP/6-31 + G(d) level[30] and the VCI method based on the many-body PEFs[36], and the experimentally measured one[10] for comparison. We also compare with the harmonic spectrum computed at the full-system MP2/aug-cc-pVDZ level (shown in Supplementary Fig. 2). The difference between the CCD/ harmonic and CCD/VQDPT2 spectra computed in this study is quite pronounced (see Fig. 2). The harmonic spectrum shows distinct blue shifts in the range of 3000–3800 cm⁻¹

compared to the VQDPT2 spectrum. In addition, the strong vibrational bands in a range of 1800–2200 cm⁻¹ in the VQDPT2 spectrum are totally absent in the harmonic spectrum. The anharmonic correction leads to a drastic change in the spectral shape. Consequently, the VQDPT2 spectrum is much closer to the experimental result compared to the one predicted by the harmonic approximation, exemplifying the crucial role of anharmonicity in the characterization of the IR spectra of protonated water clusters.

When comparing with the previous VPT2 spectrum at the B3LYP/6-31 + G(d) level, the present result shows significant improvement in predicting the experimental IR bands, confirming the importance of using the high-level theoretical treatment for both the electronic (CCD/aug-cc-pVDZ) and vibration (VQDPT2) parts. The VPT2 spectrum results in significant deviations from the experiment, showing blue-shifted bands in a low frequency range of 1200–2600 cm⁻¹ and distinctly red-shifted bands in a high frequency range of 2800–3700 cm⁻¹. These discrepancies hamper a definitive assignment of the observed bands. In stark contrast, the VQDPT2 spectrum based on fragment-based CCD/aug-cc-pVDZ agrees very well with experiment in both low and high frequency regions. Note that the correction of higher-level electronic correlation effects by employing CCSD/aug-cc-pVTZ supports our present predictions (see Supplementary Fig. 3 and Supplementary Note 3), which substantiates the effectiveness of the CCD/aug-cc-pVDZ level in computing the IR spectral features for quantitative assignments. We further emphasize that the O-H bond-stretching PESs of the $H_3O^+$ cation and the $H_2O$ molecule, as well as their intermolecular interaction PESs, calculated by B3LYP/6-31 + G(d) deviate significantly from benchmark results obtained

using a high-level wavefunction theory (CCSD(T)/aug-cc-pVQZ), while the PESs calculated by CCD/aug-cc-pVDZ are in good agreement with the benchmarks (shown in Supplementary Fig. 4). This further justifies the thesis that the vibrational band assignments benefit from high-level electronic and vibrational structure theories.

The previous VCI spectrum matches well with the present result. Some notable differences are: (1) VCI gives no signal in a range of 600–900 $cm^{-1}$ because the librational modes of $H_2O$ were excluded from the calculation. (2) The IR band shape in a range of 1700–2800 $cm^{-1}$ appears different, where VCI exhibits a strong, broad band around 1950 $cm^{-1}$ and diminishes beyond 2200 $cm^{-1}$, whereas VQDPT2 yields sharp peaks up to ~2700 $cm^{-1}$. Nevertheless, the overall agreement of the IR spectra obtained by two different theoretical approaches indicates the robustness of the calculated results.

**Band assignment**. The assignment of the spectral features associated with the proton defect is of fundamental importance. The proton-induced absorptions and the associated motions of the surface-bound $H_3O^+$ (see Supplementary Fig. 5) are characterized in detail below. One of the three frustrated rotations gives a strong peak at 943 $cm^{-1}$ ($\nu^r_{H_3O^+}$). The distinct and isolated peak calculated at 1267 $cm^{-1}$ corresponds to the umbrella vibration ($\nu^u_{H_3O^+}$), which agrees with the experimental band at 1220 $cm^{-1}$. The calculated continuous absorption occurring over the range of 1720–2100 $cm^{-1}$, corresponding to a broad band in the same region in the experiment, is due to the asymmetric O-H stretching modes of the surface-bound $H_3O^+$, which supports the assignment in the previous works[10,36]. Two asymmetric O-H stretching modes of $H_3O^+$ ($\nu^{a_1}_{H_3O^+}$ and $\nu^{a_2}_{H_3O^+}$), illustrated in Figs. 1B and 1C, are predicted at 1791 and 2035 $cm^{-1}$, respectively. Moreover, the peak at 1949 $cm^{-1}$ is attributed to a resonance state of the asymmetric O-H stretching modes of $H_3O^+$, combination tones of the frustrated rotation and umbrella vibration of $H_3O^+$, and the libration of DDA$^h$ water ($\nu^{a_1,a_2,r+u}_{H_3O^+}$). Note that the H-O-H bending of $H_3O^+$ is calculated at ~1730 $cm^{-1}$, but its intensity is too weak to be noticed. The origin of the strong band near 2220 $cm^{-1}$ and the shoulder near 2400 $cm^{-1}$ has been a matter of discussion. It was speculated that the band at 2220 $cm^{-1}$ stemmed from a combination of $H_3O^+$ bend and the frustrated rotation by comparing with a similar band in an Eigen cluster, $H^+(H_2O)_4$[10,47]. The present calculation predicts two absorption bands at 2261 and 2409 $cm^{-1}$, which correspond to the experimental signatures of ca. 2220 $cm^{-1}$ and 2400 $cm^{-1}$, respectively. The former is assigned to a resonance state of the asymmetric O-H stretching of $H_3O^+$ and the combination tones of frustrated rotation and umbrella vibration of $H_3O^+$ ($\nu^{a_1,r+u}_{H_3O^+}$, $\nu^{a_2,r+u}_{H_3O^+}$), while the latter is attributed to the symmetric O-H stretch of $H_3O^+$ ($\nu^s_{H_3O^+}$). The broad absorption around 2720 $cm^{-1}$ was rarely explored due to the weak intensity in the experimental spectrum. The present calculation offers a clear-cut assignment of the 2720 $cm^{-1}$ band to a resonance state of the asymmetric O-H stretching and a combination tone of the frustrated rotation and $H_3O^+$ bend ($\nu^{a_1,r+b}_{H_3O^+}$, $\nu^{a_2,r+b}_{H_3O^+}$). Therefore, our work reveals the origin of bands associated with the motion of the surface-bound $H_3O^+$. The assignments are summarized in Supplementary Table 1.

Let us now focus on IR bands of neutral water molecules. The experimental spectrum gives broad features in a low-frequency range from 600 to 1000 $cm^{-1}$, which was assigned to the librational motion of neutral water molecules[10]. The present calculation supports the assignment, yielding IR bands of the librational motion in the same range. The three predicted absorption peaks (696, 720, and 746 $cm^{-1}$), occurring below 750 $cm^{-1}$, arise from the libration of three neighboring DDA$^h$-type neutral water molecules around the surface-bound $H_3O^+$ ion in the single hydrogen-bond acceptor configurations. The remaining bands in this region mainly correspond to the libration of AAD-type water molecules far from $H_3O^+$. The strongest peak calculated at 855 $cm^{-1}$ agrees well with the experimental peak around 840 $cm^{-1}$. The detailed assignments of the librational bands are summarized in Supplementary Table 2, and the vibrational motion is illustrated in Supplementary Fig. 6.

The bending vibrations of neutral water molecules are predicted to be around 1600 $cm^{-1}$, in good agreement with a sharp band observed at 1620 $cm^{-1}$ in the experiment. The broad envelope from 3000 to 3600 $cm^{-1}$, with sequential peaks denoted by letters (**a**–**i**), is associated with the O-H stretching motions of the neutral water molecules. These features have attracted increased attention, because they serve as a useful marker to characterize the hydrogen bond network in various systems[34]. Yu and Bomwan[36] assigned some of these spectral signatures, tracing them to specific types of water molecules through the one-to-one correspondence with the VCI spectrum. We have performed a more comprehensive analysis on the VQDPT2 spectrum and reconfirmed most of their assignments. The assignments of neutral water O-H stretching bands are shown in Fig. 3 and the details are given in the Supplementary Note 1 and Supplementary Table 3.

**Two types of AADD water**. It is noticeable that the O-H stretching frequency of the AADD$^s$-type water molecules is significantly higher than that of the AADD$^i$-type water molecule (features **b** and **f**, respectively, in the upper panel of Fig. 3), which suggests two distinct structures of four-coordinated water molecules. The structural comparison of the AADD$^i$ and AADD$^s$ water molecules in the $H^+(H_2O)_{21}$ cluster, with reference to the ice Ih[48] and liquid water[42], is summarized in Table 1. The major difference between the interior and surface four-coordinated water of $H^+(H_2O)_{21}$ cluster lies in the angle of the hydrogen-bonds. The angles of the hydrogen-bonds formed between the AADD$^i$-type water molecule and its neighbors are all near 177°, as shown in the inset of the upper panel of Fig. 3, which is almost the same as the average hydrogen-bond angle (178°) in ice Ih. These strong hydrogen bonds of the AADD$^i$-type water molecule in the ice-like structure accounts for the relatively low O-H stretching frequency. In contrast, the average hydrogen bonds formed between the AADD$^s$-type water molecules and its partners are distorted (see the inset of Fig. 3), exhibiting the structural property similar to liquid water. The average hydrogen-bond angles of liquid-like AADD$^s$ water molecules is 162° in the $H^+(H_2O)_{21}$ cluster, which is very close to the average angle of 158° in liquid water, indicating that the AADD$^s$ water molecules are less confined, and its hydrogen-bond strength with the partners is weaker than that of the AADD$^i$ water molecule. These imperfect, and therefore weaker hydrogen bonds, are responsible for the high O-H vibrational frequency of more flexible AADD$^s$ water molecules under less confined environment, as compared to the AADD$^i$ water molecule. The difference of the two types of AADD water molecule in the $H^+(H_2O)_{21}$ cluster is also addressed in comparison with the experimentally observed bulk spectra of ice Ih[49] and liquid water[50], as shown in Supplementary Fig. 7. The O-H stretching frequency of liquid water shows a clear blue shift by ~180 $cm^{-1}$ with reference to ice Ih, which is in good agreement with the present calculation (shifted by ~165–185 $cm^{-1}$) and further proves the existence of two types of AADD water molecule in the $H^+(H_2O)_{21}$ cluster. Our study demonstrates that there are

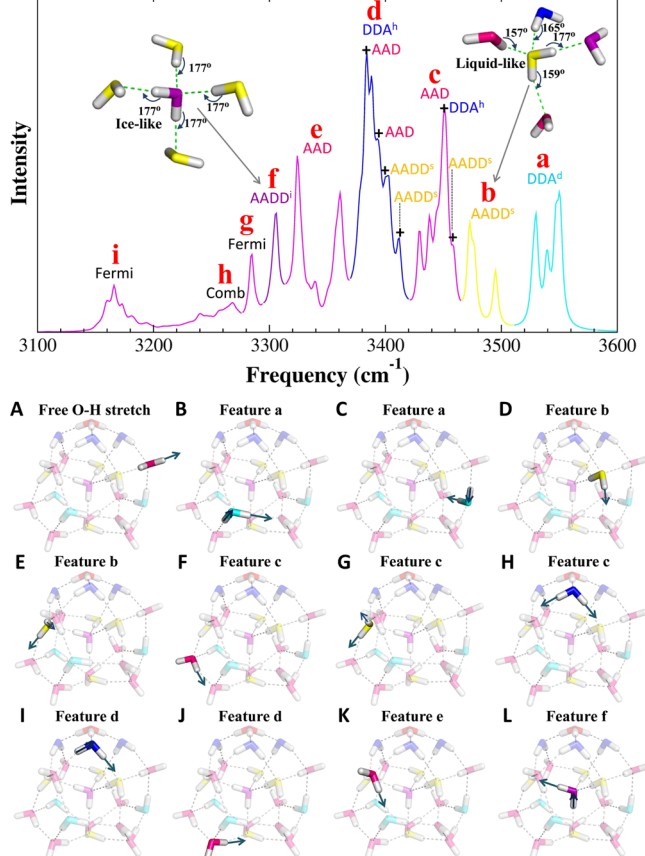

**Fig. 3 Assignments of O-H stretching features of neutral water molecules.** (Upper panel) Enlarged view of the VQDPT2 spectrum in the range of O-H stretching of neutral water molecules. The assignments are indicated in colors and labels according to the type of corresponding water molecules. Notations: Fermi resonance (Fermi), combination band (Comb), mixed vibration with different type of water (+). (Lower panel) The O-H stretching motions of the neutral water molecules. (A) Free O-H stretch of the AAD-type water molecule. (B), (C) Two DDA-type water molecules in feature (**a**). (D), (E) The AADD$^s$-type water molecules in feature (**b**). (F), (G), (H) The AAD, AADD$^s$, and DDA$^h$-type water molecules in feature (**c**). (I), (J) The DDA$^h$ and AAD-type water molecules in feature (**d**). (K) The AAD-type water molecule in feature (**e**). (L) The AADD$^i$-type water molecule in feature (**f**).

**Table 1 The structural comparison of the simulated liquid water, ice Ih, and the H$^+$(H$_2$O)$_{21}$ cluster ($d$ in angstrom, and $\angle$ in degree). The results for the H$^+$(H$_2$O)$_{21}$ cluster are average values over 4 cluster structures shown in Fig. 5.**

| | Liquid water$^e$ | Ice Ih$^f$ | H$^+$(H$_2$O)$_{21}$ cluster | |
| --- | --- | --- | --- | --- |
| | | | AADD$^i$ | AADD$^s$ |
| $d_{OH}{}^a$ | 0.97 | 0.98 | 0.98 | 0.97 |
| $\angle HOH^b$ | 104 | 106 | 105 | 105 |
| $d_{O-O}{}^c$ | 2.93 | 2.76 | 2.77 | 2.79 |
| $\angle O_D - H_D..O_A{}^d$ | 158 | 178 | 177 | 162 |

$^a$The average O-H bond length of the water molecule.
$^b$The average H-O-H angle of the water molecule.
$^c$The average hydrogen-bond length in terms of the oxygen–oxygen distance between two hydrogen-bonded water molecules.
$^d$The average hydrogen-bond angle formed between the O$_D$-H$_D$ bond and the H$_D$..O$_A$ direction.
$^e$The data were calculated at the fragment-based CCD/aug-cc-pVDZ level from ref. [42].
$^f$The data were calculated at the fragment-based MP2/aug-cc-pVDZ level from ref. [48].

two kinds of four-coordinated water molecules in the H$^+$(H$_2$O)$_{21}$ cluster based on direct comparison between the experimental IR spectrum and high-level wavefunction theory calculations. The internal and surface four-coordinated AADD water molecules correspond to the ice-like and liquid-like water, respectively, as indicated by their differences in local tetrahedral structure, hydrogen-bond strengths, and vibrational spectral signatures. This can provide a bottom-up framework for understanding the structural differences at the molecular level between fully coordinated, bulk-like water and interfacial water at the water/solid or water/vapor interfaces[51].

**Intermolecular couplings of water molecules.** Given an increasing interest in the relaxation of O-H stretching vibration in caged water clusters[52], let us address the effect of the water–water couplings on the calculated spectrum. The intermolecular coupling between the water molecules were taken into account in VQDPT2 calculations by including bi-linear coupling terms in the PES. Note that the water–water coupling was excluded in the previous work[36]. The VQDPT2 spectrum excluding the water–water couplings (except for DDA$^h$-DDA$^h$ so as to retain the inter-molecular couplings of the H$_3$O$^+$(H$_2$O)$_3$ moiety) is shown in Fig. 4A for the O-H stretching region, and in Supplementary Fig. 8 for the lower frequency region. The two spectra with and without the water–water couplings give major peaks in similar positions, and thus the overall appearance looks similar. Nonetheless, the presence of the water–water coupling generally makes the spectrum more broadened and widespread. For example, the O-H stretching band in a range of 3100–3400 cm$^{-1}$ exhibits noticeable differences. In the fully coupled model, the peaks calculated at 3284 and 3166 cm$^{-1}$ (denoted as **g** and **i**, respectively), manifest intra-molecular Fermi resonance of an AAD-type water molecule between an overtone of the bending mode (No. 100) and a fundamental of the O-H stretching mode (No. 101). Interestingly, the lower frequency component is further resonant with an overtone of the bending mode (No. 127) of a neighboring AAD-type water molecule. The vibrational modes, the resonance diagram, and the component of vibrational wavefunctions are shown in Fig. 4B. It is notable that the coupling constant between modes 100 and 127 (calculated as 12 cm$^{-1}$) is not particularly large compared to others; for example, the coupling constants of mode 100 with bending modes of DDA$^d$- and AADD$^s$-type water molecules in the nearest neighbors are obtained as 14 and 16 cm$^{-1}$, respectively. Instead, the state mixing is induced by the match of frequencies, where the fundamental excitations of the two AAD-type water molecules are obtained as 1605 and 1609 cm$^{-1}$, whereas those of DDA$^d$- and AADD$^s$-type water molecules are higher in frequency at 1640 and 1651 cm$^{-1}$. The result implies a novel relaxation pathway of the O-H stretching excitation energy through bending overtone states mediated by the vibrational resonance[53]. These peaks are observed around 3230 and 3110 cm$^{-1}$ in the experiment.

**Theoretical IR spectra of other structures.** As all the key bands in the H$^+$(H$_2$O)$_{21}$ IR spectrum have been assigned to the particular network sites based on the calculation of the structure in its most stable form, it is necessary to compare the computed spectrum to those of other minimum energy structures. To this end, three reported alternative stable structures of H$^+$(H$_2$O)$_{21}$ cluster[33] were also computed in the present study. The Cartesian coordinates of these structures are provided in the Supplementary Note 4. For ease of comparison, the four minimum energy structures considered in this study are displayed in Fig. 5 (note that a1 is the

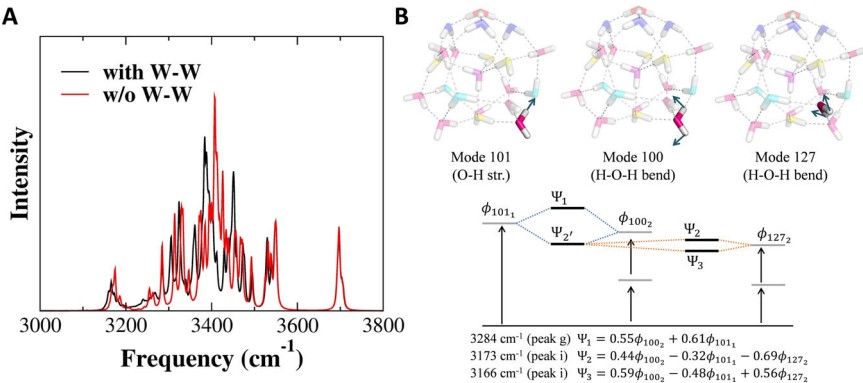

**Fig. 4 On the effect of water–water (W–W) interactions. A** Comparison of the IR spectra obtained by VQDPT2 with and without the harmonic coupling between water molecules in the PES. Note that the $H_3O^+(H_2O)_3$ moiety has all inter-molecular couplings included in both cases. **B** The vibrational modes, the resonance diagram, and the component of vibrational wavefunctions for the intra- and inter-molecular Fermi resonance of the AAD-type water molecules.

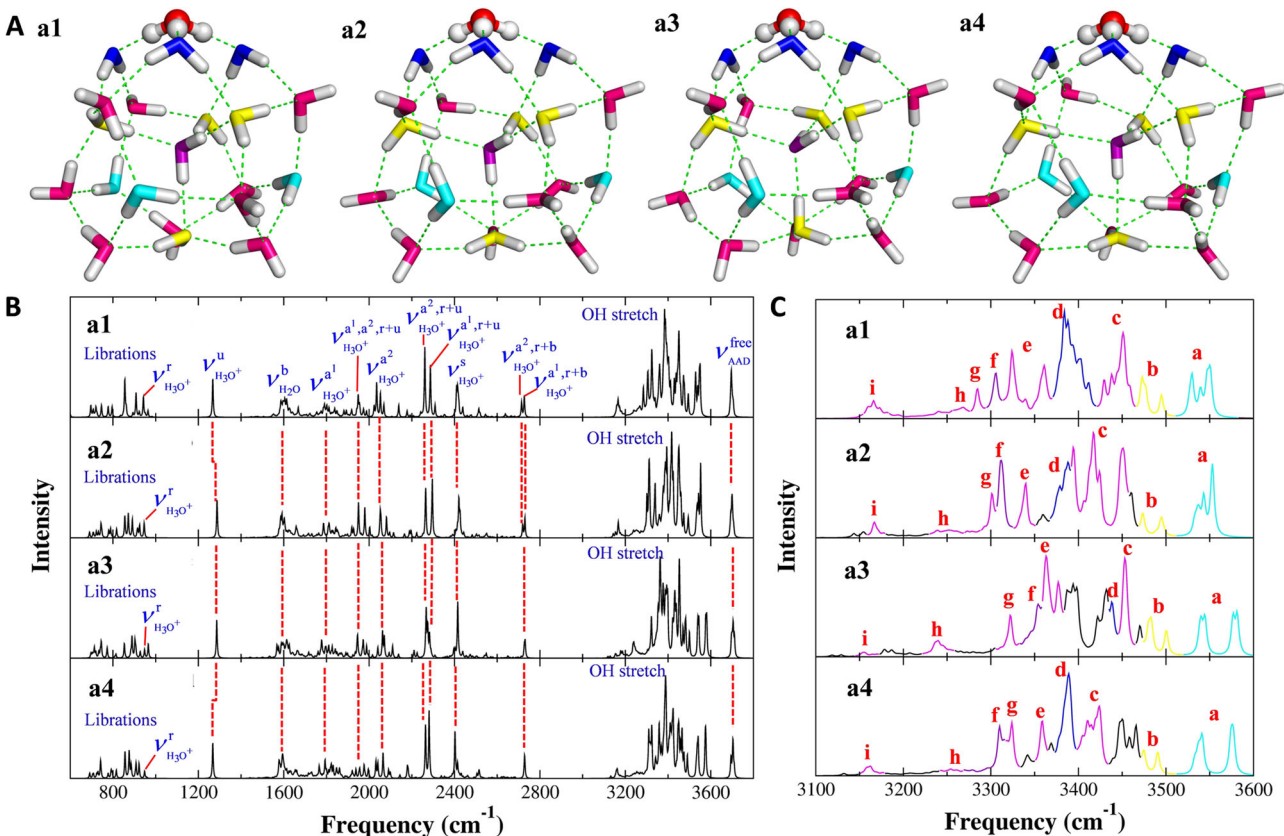

**Fig. 5 Different minimum energy structures and corresponding IR spectra. A** Overview of the four minimum energy structures of $H^+(H_2O)_{21}$ reported in Hodges and Wales's study[33], and (**B**), (**C**) the corresponding VQDPT2 IR spectra computed at the fragment-based CCD/aug-cc-pVDZ level. The enlarged view of the O-H stretching region of the spectra in (**C**) is color-coded according to the type of the corresponding water molecules.

structure discussed above) along with the corresponding VQDPT2/CCD predictions. The numbers of the water molecules in the same types from the four structures are equal, and their relative positions are almost identical. The $H_3O^+$ and three DDA[h]-type water molecules have the same hydrogen-bond network in the four configurations (see Fig. 5A). Therefore, the key bands associated with the proton defect ranging from 1200 to 2800 cm$^{-1}$ in the calculated IR spectra show a very similar pattern for the four structures (see Fig. 5B). The major difference among the four structures lies in the different orientations of individual water molecules in AAD, DDA[d], AADD[s], and AADD[i] types (see

Fig. 5A), leading to different hydrogen-bond partners for water molecules at the same positions among the four structures. The perturbation of hydrogen-bond network of individual water molecules results in a slight rearrangement of the spectrum in the O-H stretching region from 3100 to 3600 cm$^{-1}$ (see Fig. 5C), which is sensitive to the hydrogen-bond structure. Although the shapes of the O-H stretch-induced absorptions are slightly different for the four structures, all of them have the characteristic features as those in the structure a1, and the same features derived from the specific type of water appear at very close positions. A few individual features unique in the structures a2, a3 and a4 are

described in the Supplementary Note 2. In essence, this comparative study emphasizes that different minimum energy structures are considered to sufficiently warrant the correct assignments of the distinct bands.

## Discussion

We have developed a new protocol to compute the IR spectrum of molecular clusters based on the combination of EE-GMF and VQDPT2 for treating the electronic and vibrational problems, respectively. In the EE-GMF method, all monomers and dimers are calculated by the ab initio electronic structure calculations with the electrostatic embedding scheme. This scheme, in which the environmental effects are incorporated by surrounding atomic point charges, accounts for the electronic polarization and hydrogen-bond cooperativity effects, thereby making the truncation after the dimer terms far more accurate than a simple summation of bare monomer/dimer energies. VQDPT2 treats the strong interaction among quasi-degenerate states by VCI, and the weak interaction with many, non-degenerate states by the second-order perturbation theory. Unlike the regular perturbation theory, VQDPT2 is capable of describing resonance states without divergence while keeping the cost-efficiency and the scalability to many-mode systems. Furthermore, we employ vibrational coordinates localized to each molecule and represent the PES in terms of "intra"-molecular anharmonicity and "inter"-molecular harmonic coupling. The PES generated by EE-GMF is used for VQDPT2 calculations. The method is an ideal combination to compute the vibrational spectrum of molecular clusters, exploiting the locality of electronic and vibrational motions.

These methodological advances signify the nearly complete assignment of the IR spectral features of the $H^+(H_2O)_{21}$ cluster, 17 years after it has been experimentally measured. The calculated spectrum not only reproduces the well-defined structures for the bands previously assigned, but also provides definitive structural proof for the clarification of the previously controversial and unclear band assignments of proton motions. We emphasize that the revelation of the IR band assignments has a profound impact on the understanding of molecular structures in various systems. The precise assignments of the proton defect band in a hydrogen bonded network carve a path for addressing several open questions related to the nature of proton speciation in water. The site-specific analysis of the water O-H stretching region reveals distinct structures for the internal ice-like and surface liquid-like four-coordinated water molecules that are the cornerstone of understanding the local structure of water in diverse environments; for example, the water/air or solid interface, water clusters and droplets in amorphous polymer, and so on.

The present calculation is complementary with the previous VCI calculation based on the many-body PEFs by Yu and Bowman. On one hand, the PEFs of $H^+(H_2O)_n$ ($n = 1–4$) and $(H_2O)_n$ ($n = 1–3$) were derived at the mixed high electronic structure levels of CC and MP2, but they were simply summed to construct the PES of $H^+(H_2O)_{21}$, whereas the PES in our calculation is computed for $H^+(H_2O)_{21}$ by the fragment-based CC in an electrostatically embedding scheme at the level of CCD/aug-cc-pVDZ. On the other hand, VCI was carried out for a $H_3O^+$ $(H_2O)_3$ moiety in 15 dimensions incorporating ~140,000 of VCI states and other water molecules in 3 dimensions, whereas VQDPT2 was performed for $H^+(H_2O)_{21}$ in 89 dimensions incorporating ~1000 of quasi-degenerate states by VCI and hundreds of millions of non-degenerate states by perturbation. There are multiple measures on the level accuracy, and the two approaches are complementary with each other. Nonetheless, the resulting IR spectra exhibit an overall agreement, which

substantiates the robustness of the theory even for such a complex system as $H^+(H_2O)_{21}$.

Although the present calculation predicted the IR peak position in good match with the experiment, the agreement of the intensity and line-shape is less sufficient, in particular, in the range of 1700–2700 cm$^{-1}$. This is primarily because the calculated spectrum was constructed by simply augmenting the peak position and intensity using Lorentz functions of constant FWHM (5 cm$^{-1}$). The procedure is valid when the excited state has long lifetime. However, the broad, line-shape observed in the experiment indicates fast dynamics of the proton defect and the vibrational mode mixing after the excitation of O-H stretching vibration of $H_3O^+$. We also found an indication of the intermolecular energy relaxation pathway of the O-H stretching excitation of neutral water molecules via H-O-H bending overtones. With the advent of the experimental techniques (IR–IR hole burning, 2D-IR, etc.), revealing the fast dynamics of proton defect and water molecules is intriguing. Further theoretical improvement is needed to extend the framework to time-dependent quantum theory as well as to generate a more accurate PES for quantum dynamics, which will be the scope of future works.

## Methods

**The electrostatically embedded generalized molecular fractionation (EE-GMF) method.** Fragment-based quantum chemical methods[54,55], in which a large system is decomposed into small, tractable pieces more affordable to electronic structure calculations, have been proposed as an effective way to sidestep the non-linear scaling of standard quantum mechanical (QM) computational cost with respect to the system size. The electrostatically embedded generalized molecular fractionation (EE-GMF) method was developed in our group to treat large-sized molecular clusters[41,42,45,56–59]. As a fragment-based quantum chemical method, the EE-GMF approach has been elaborated in a series of our recent publications[41,42,56,57,59], and thus we only give a brief description here. The EE-GMF approach was developed for specifically dealing with molecular clusters, in which each molecule could be assigned as a single fragment without cutting the chemical bonds. Then each fragment, with the remaining system represented by the background charges, could be feasibly treated at diverse ab initio levels. The interactions between two fragments that are spatially in close contact have important contributions to the energetic properties of the system, and hence are also calculated by QM, while the long-range electrostatic interactions are approximated using the classical Coulomb interactions for efficiency. Therefore, according to the EE-GMF scheme, the total energy ($E$) of the molecular cluster can be expressed by,

$$E_{molecular\ cluster}^{EE-GMF} = \sum_{i=1}^{N} \tilde{E}_i + \sum_{i=1}^{N-1} \sum_{\substack{j=i+1 \\ |R_{ij}| \leq \lambda}}^{N} (\tilde{E}_{ij} - \tilde{E}_i - \tilde{E}_j) - \sum_{i=1}^{N-1} \sum_{\substack{j=i+1 \\ |R_{ij}| > \lambda}}^{N} \sum_{m \in i} \sum_{n \in j} \frac{q_{m(i)} q_{n(j)}}{R_{m(i)n(j)}} \quad (1)$$

where $N$ is the number of the fragments in the molecular cluster, $\tilde{E}_i$ denotes the self-energy of the fragment $i$ along with the interaction energy between the fragment and background charges of the rest of the atoms in the system, the second term in Eq. (1) denotes the two-body QM interaction energies between dimer $ij$ (which is composed of fragments $i$ and $j$), when the distance $R_{ij}$ between fragments $i$ and $j$ is less than or equal to a predefined distance threshold $\lambda$, and $q_{n(j)}$ represents the atomic charge of the $n$th atom in the $j$th fragment. The last term in Eq. (1) deducts the doubly counted electrostatic interactions between distant fragment pairs (outside the distance threshold $\lambda$), because those interactions are already taken into account in each fragment QM calculations with the electrostatic embedding scheme. Here in this study, each water molecule and hydronium ion $H_3O^+$ were assigned as individual fragments, and all the two-body interactions between any two fragments were calculated by QM (i.e., $\lambda$ is chosen to be sufficiently large to cover all the two-body QM interactions). The higher-order many-body interactions are implicitly incorporated in the electrostatic embedding scheme. In this case, Eq. (1) is simplified to $E_{molecular\ cluster}^{EE-GMF} = \sum_{i=1}^{N} \tilde{E}_i + \sum_{i=1}^{N-1} \sum_{j=i+1}^{N} (\tilde{E}_{ij} - \tilde{E}_i - \tilde{E}_j)$, which becomes similar to the electrostatically embedded many-body expansion (EE-MB) method proposed by Dahlke and Truhlar[60,61].

With the EE-GMF fragmentation method, all of the monomers and dimers were explicitly treated through the standard ab initio calculations with the electrostatically embedding scheme to account for the environmental effect, which ensures the electronic polarization and hydrogen-bond cooperativity to be properly taken into consideration. Through the monomer and dimer calculations, one- and two-body electronic Coulomb, exchange, and correlation interactions are treated nearly exactly at the CCD level. By means of the electrostatic embedding approach,

three-body and all higher-order many-body Coulomb interactions are also included implicitly. It is the electrostatic embedding scheme that renders the many-body expansion quickly convergent, and the truncation after the dimer QM interactions sufficiently accurate. The atomic charges utilized for the embedding field were obtained from the SPCFW[62] water model and the electrostatic potential fitting at the HF/aug-cc-pVDZ level for the protonated water $H_3O^+$. The first and second derivatives of the total energy with respect to the nuclear coordinates, i.e., the atomic forces and Hessian matrix, can be calculated analytically[42,57,59,63], which were utilized for geometry optimization and normal mode analysis. The quasi-Newton algorithm was adopted for $H^+(H_2O)_{21}$ geometry optimization from a given initial structure, and the BFGS procedure was used to update the Hessian matrix during the optimization procedure. The convergence criterion of the maximum atomic force was set to 0.001 Hartree/Bohr.

The dipole moment of the molecular cluster ($\mu$) can also be obtained based on the EE-GMF scheme as,

$$\mu_{molecular\ cluster}^{EE-GMF} = \sum_{i=1}^{N} \mu_i + \sum_{i=1}^{N-1} \sum_{\substack{j=i+1 \\ |\mathbf{R}_{ij}| \le \lambda}}^{N} (\mu_{ij} - \mu_i - \mu_j) \quad (2)$$

where $\mu_i$ is the dipole moment of fragment $i$. In this work, all the two-body corrections on the dipole moment of the entire molecular cluster between any two fragments are calculated by QM (i.e., $\lambda \to \infty$, and $\mu_{molecular\ cluster}^{EE-GMF} = \sum_{i=1}^{N} \mu_i + \sum_{i=1}^{N-1} \sum_{j=i+1}^{N} (\mu_{ij} - \mu_i - \mu_j)$). The derivative of the dipole moment with respect to the normal coordinates can also be derived to compute the IR intensity[59,63].

### The second-order vibrational quasi-degenerate perturbation theory (VQDPT2).
VQDPT2[43,44] is an efficient method to solve the vibrational Schrödinger equation (VSE). The vibrational Hamiltonian reads in terms of mass-weighted, rectilinear vibrational coordinates, $\{Q_i\}$, as

$$\hat{H}_v = -\frac{1}{2} \sum_{i=1}^{f} \frac{\partial^2}{\partial Q_i^2} + V(\mathbf{Q}) \quad (3)$$

where $f$ is the number of vibrational degrees of freedom and $V$ is the potential energy surface (PES) of a system. The vibrational self-consistent field (VSCF) wavefunction is the starting point of the calculation,

$$|\Phi_{\mathbf{n}}^{VSCF}\rangle = \prod_{i=1}^{f} |\phi_{n_i}^{(i)}(Q_i)\rangle \quad (4)$$

where $n$ denotes the quantum number of a target vibrational state. The one-mode functions are obtained by solving the VSCF equation,

$$\left[ -\frac{1}{2} \frac{\partial^2}{\partial Q_i^2} + \left\langle \prod_{i' \ne i} \phi_{n_{i'}}^{(i')} |V| \prod_{i' \ne i} \phi_{n_{i'}}^{(i')} \right\rangle \right] \phi_{n_i}^{(i)} = \varepsilon_{n_i} \phi_{n_i}^{(i)} \quad (5)$$

VQDPT2 improves the VSCF solution using the second-order quasi-degenerate perturbation theory. We divide the Hilbert space into a $P$ space spanned by VSCF configuration functions, $\{\Phi_{\mathbf{p}}^{VSCF}\}$, in which the components are energetically quasi-degenerate to target states, and a complimentary $Q$ space, $\{\Phi_{\mathbf{q}}^{VSCF}\}$. The effective Hamiltonian is written up to the second order as,

$$\left( H_{eff}^{(0+1)} \right)_{\mathbf{PP'}} = \left\langle \Phi_{\mathbf{P}}^{VSCF} | \hat{H}_v | \Phi_{\mathbf{P'}}^{VSCF} \right\rangle \quad (6)$$

$$\left( H_{eff}^{(2)} \right)_{\mathbf{PP'}} = \sum_{\mathbf{q} \ne \mathbf{p}} \frac{\left\langle \Phi_{\mathbf{P}}^{VSCF} | \hat{H}_v | \Phi_{\mathbf{q}}^{VSCF} \right\rangle \left\langle \Phi_{\mathbf{q}}^{VSCF} | \hat{H}_v | \Phi_{\mathbf{P'}}^{VSCF} \right\rangle}{2} \left( \frac{1}{E_{\mathbf{P}}^{(0)} - E_{\mathbf{q}}^{(0)}} + \frac{1}{E_{\mathbf{P'}}^{(0)} - E_{\mathbf{q}}^{(0)}} \right) \quad (7)$$

where $E_{\mathbf{P}}^{(0)}$ is the zero-th order energy defined as,

$$E_{\mathbf{P}}^{(0)} = \sum_i \varepsilon_{p_i} \quad (8)$$

The diagonalization of the effective Hamiltonian yields the VQDPT2 energy and wavefunctions. The $P$ and $Q$ space is constructed using two control parameters, $N_{gen}$ and $\lambda_{max}$, for a target vibrational state of interest, $n$. VSCF configurations that are quasi-degenerate to $n$ are searched in a configuration space $\{s\}$ defined by $\lambda_{max}$ as,

$$\lambda_{\mathbf{sn}} = \sum_{i=1}^{f} |s_i - n_i| \le \lambda_{max} \quad (9)$$

The quasi-degenerate configurations found in the search are denoted $\{\mathbf{p}^{(1)}\}$. Then, the same search is carried out for each configuration of $\{\mathbf{p}^{(1)}\}$ to find the second generation of quasi-degenerate configurations, $\{\mathbf{p}^{(2)}\}$. The process is repeated $N_{gen}$ times to obtain the $P$ space configurations,

$$P = \{\mathbf{n}\} + \{\mathbf{p}^{(1)}\} + \{\mathbf{p}^{(2)}\} + \cdots + \{\mathbf{p}^{(N_{gen})}\} = \{\mathbf{p}^m | m = 1, 2, \cdots, N_P\} \quad (10)$$

The $Q$ space is constructed by selecting configurations, q, that satisfies the following condition,

$$\lambda_{\mathbf{qp}^m} = \sum_{i=1}^{f} |q_i - p_i^m| \le \lambda_{max} \quad (11)$$

Note that the $Q$ space configurations are non-degenerate with any of the $P$ space configurations. The energy differences in the denominator of Eq. (7) are finite, and thus VQDPT2 is free of a divergence problem. The relation of VQDPT2 with other vibrational methods and the vibrational calculations based on local coordinates are described in the Supplementary Methods.

**Computational details.** The protonated $H^+(H_2O)_{21}$ cluster with an Eigen state hydrated hydronium cation sitting on the surface of the water cage was utilized in this study. The four most stable conformers of $H^+(H_2O)_{21}$ were obtained from Hodges and Wales's study[33], and used as the initial structures for geometry optimization. The Coupled-Cluster Doubles (CCD) theory with the aug-cc-pVDZ basis set was applied for geometry optimization and harmonic vibrational calculation by using the EE-GMF method. All QM calculations were carried out with the Gaussian16 program[64].

The anharmonic vibrational calculations were carried out in terms of coordinates localized to each molecule ($H_3O^+$ and $H_2O$). We employed the nine- and four-highest frequency modes of $H_3O^+$ and $H_2O$, respectively. Thus, 89 out of 192 coordinates were set to be active. The harmonic frequencies of the local coordinates (and their counterparts in normal coordinates) are listed in Supplementary Table 4. In Eq. (8) of the Supplementary Methods, the intra-molecular PES was generated at the anharmonic level up to the three-mode representation (3MR) by the multi-resolution method[65] combining the QFF[66] and the grid PES[67]. The one-mode representation (1MR)-PES was a grid PES with 11 grid points for all terms, while the two-mode representation (2MR) and 3MR coupling terms with mode coupling strength (MCS)[68] larger than 1.0 and 10.0 $cm^{-1}$ were obtained by a grid PES with 7 and 5 grid points, respectively. Other weaker terms were represented by QFF. The generation of QFF and grid PES required 981 points of gradient and 11,263 points of energy, respectively, which were computed by the EE-GMF method based on the level of CCD/aug-cc-pVDZ. In addition, the 1MR grid PES was also calculated at the CCSD/aug-cc-pVTZ level for more accurate description of the electronic correlation effect. The inter-molecular, harmonic coupling was obtained from the Hessian matrix. Finally, the VQDPT2 calculations[43,44] were performed with $N_{gen} = 3$ and $\lambda_{max} = 4$. The target vibrational states were set to fundamental excitations. The largest $P$ and $Q$ spaces incorporated 920 and 333 million VSCF configurations, respectively. The IR intensities were computed using the dipole moment surfaces obtained from the same grid points as the grid PES. All anharmonic vibrational calculations were carried out using the SINDO program[69].

## Data availability
All of the input and output files for quantum mechanical and vibrational calculations underlying the conclusions of this work are archived in https://github.com/jinfeng-data/H-H2O-21.

## Code availability
The Gaussian16 program used to perform the quantum mechanical calculations in this work is commercially available (http://www.gaussian.com/). All anharmonic vibrational calculations were carried out using the SINDO program, which is publicly available (https://tms.riken.jp/en/research/software/sindo/). The MP2 harmonic analysis was performed with the NWChem suite of codes. The EE-GMF codes are archived in https://github.com/jinfeng-data/EE-GMF[70].

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

## Acknowledgements
X.H. and J.L. were supported by the National Natural Science Foundation of China (Grant Nos. 21703289, 21922301, 21761132022, and 21673074), the National Key R&D Program of China (Grant Nos. 2016YFA0501700, 2019YFA090402, and 2019YFA0905201), Shanghai Municipal Natural Science Foundation (Grant No. 18ZR1412600), "Double First-Class" University project (CPU2018GY09), the Fundamental Research Funds for China Pharmaceutical University (2632019FY01), and the Fundamental Research Funds for the Central Universities. K.Y. was supported by JSPS KAKENHI (Grant No. 20H02701). S.S.X. acknowledges support from the US Department of Energy, Office of Science, Office of Basic Energy Sciences, Division of Chemical Sciences, Geosciences and Biosciences at Pacific Northwest National Laboratory (PNNL). PNNL is a multiprogram national laboratory operated for DOE by Battelle. This research also used resources of the National Energy Research Scientific Computing Center, which is supported by the Office of Science of the U.S. Department of Energy under Contract No. DE-AC02-05CH11231. The Supercomputer Centers of East China Normal University (ECNU Multifunctional Platform for Innovation 001) and China Pharmaceutical University are acknowledged for providing computer resources. We also thank Prof. Mark A. Johnson for providing the experimental IR spectral data and many helpful discussions.

## Author contributions
X.H. and K.Y. conceived of the project. X.H., J.L., and K.Y. designed the calculations, and J.L., J.Y., X.H., K.Y., and S.S.X. performed the calculations. J.L., X.H., K.Y., S.S.X., J.Y., and X.C.Z. analyzed the data. J.L. wrote the paper, and X.H., K.Y., S.S.X., X.C.Z., and J.Y. edited the paper.

## Competing interests
The authors declare no competing interests.
