## [Peer Review File · Nature Communications]

Towards complete assignment of the infrared spectrum of the protonated water cluster $\text{H}^+(\text{H}_2\text{O})_{21}$Reviewers' comments:

Reviewer #1 (Remarks to the Author):

This MS presents a tentative assignment of the $\text{H}^+(\text{H}_2\text{O})_{21}$ water cluster IR spectrum. This system, and related ones of smaller dimension, has been the topic of several papers in the past, both computational and experimental ones, mainly published in specialized physical chemistry journals. This MS does not present any new experiment and it focuses on the assignment of a previous one. To reach this goal, the authors develop a PES and employ a method already present in the literature since 2008, which is called Second-order Vibrational Quasi-Degenerate Perturbation Theory (VQDPT2).

I report here below my specific considerations, before drawing my conclusions.

1) My main concern is about the methodological approach. In the past, perturbative approaches have been shown to break down. In a 2017 paper (J. Phys. Chem. A 2017, 121, 3056–3070) it has been reported about VPT2 (which is very similar to the present approach) applied to protonated water clusters and in trying to reproduce Ref.14 that "It can be difficult to decide which modes are described well in VPT2 and which ones are not.While the harmonic frequencies do not differ by more than ca. 100 cm^{-1} between different methods, the anharmonic ones are irreproducible and vary erratically between [VPT2 and others] methods....The huge discrepancy between methods and basis sets suggests that the anharmonic frequencies of these three [proton] modes are affected by the inherent instability of perturbation methods, and should be better replaced by their harmonic counterparts". In another publication, the VPT2 methods have been shown to differ from more accurate approaches such as MCTDH and DMC by more than 500 wavenumbers in the Zundel cation, which is the smallest protonated water cluster. In a 2019 paper (J. Phys. Chem. A 2019, 123, 1399–1409), it has been reported that "The VPT2 method is comparatively cheaper for vibrational analysis, and it has been applied to the spectrum calculation of different sizes of protonated water cluster.^{10,11,31} Though many successes have been achieved using this method, it cannot fully deal with a very anharmonic system and thus cannot provide explanations to some important regions in spectra.¹".

The authors perform their calculations with VQDPT2, which for some aspects suffers from the same limitations of VPT2 (perturbative approach), and which has been introduced for the first time in 2008. The method is very briefly recalled in the SI, without details about its accuracy performance. In the literature, the accuracy of VQDPT2 has been compared with VCI on smaller protonated clusters but not with other methods. This would have benefited the broad readership of Nat. Commun. This would also have been highly desirable, given the citations that I reported in the previous paragraph. At the light of those considerations, the authors did not offer in their MS any method to compare with using the same PES to assess their results, at least at a reduced dimension. Instead, they prefer to accept that if the calculations are in some sort of agreement with the experiments, then they are accurate. I do not agree with this approach, since compensation of errors are quite common in ab initio spectroscopy, where quite often harmonic results have shown to be more accurate than anharmonic ones on some systems.

In addition, I cannot see enough novelty in their work since the VPT2 protonated water spectrum has been already published in 2011 by one of the authors. More recently, VCI has been employed on a more accurate PES (vide infra). For these reasons, I consider this work mainly of incremental type and with a reduced novelty in the field.

2) The methodological approach strongly depends on the perturbative application point. For these reasons, when a perturbative approach is adopted for spectroscopic calculations, an alternative approach based on MD is employed to check how relevant other geometry minima are in the experimental spectroscopic signal. This may include ab initio molecular dynamics, path integral molecular dynamics and semiclassical molecular dynamics. The authors try to partially remedy to this limitation by looking at 4 other similar minima. Clearly these calculations cannot be considered representative of the thousands of possible minima. Actually, the experiments have provided only indirect evidence that the protonated water is the one at the top of the cluster. It would have been much more interesting if the authors had shown that by means of a dynamical approach that indeed would have shown that the predominant spectroscopic signal is originated from these

configurations. Instead, the authors enforced these few configurations into their perturbative approach.

3) At pg.4, the authors write: "Yu and Bowman(40) have shown that the higher level, vibrational configuration interaction (VCI) method achieves a significant improvement over VPT2. Nevertheless, in their VCI calculation, the potential energy surface (PES) of $H+(H_2O)_{21}$ (41) was represented as a sum of potential energy functions for small clusters, $H+(H_2O)_n$ ($n = 1 - 4$) and $(H_2O)_n$, ($n = 1 - 3$), without directly accounting for the electronic structure of the full cluster. Therefore, a more sophisticated quantum treatment of both the electronic and vibrational structure of the $H+(H_2O)_{21}$ cluster is required to produce computed spectra to accurately assign the experimental spectral features."

I do agree that VCI is more accurate than VPT2 but I do not agree that the previous PES and method (VCI) is less accurate than the one employed in the MS. Present calculations are at the level of CCD/aug-cc-pVDZ theory, while previous PES is at the level of CCSD(T)-F12/aVQZ (see J. Phys. Chem. A 2019, 123, 1399–1409), which is by far more accurate both in the ab initio method (CCSD(T) vs CCD) and basis set (aVQZ vs aug-cc-pVDZ). Also, the previous PES includes 2 and 3 body interactions (and 4 body interactions are calculated at the level of MP2/aVTZ) for all degrees of freedom and the approach has proved to be basically equivalent to the full-dimensional (full-body) PES. Conversely, the PES of this work, which is the only methodological novelty of the MS, accounts for 1 and 2 body interactions only for the chosen degrees of freedom (i.e. 89 out of 186), while the higher order interactions are considered as a standard typical MD force field. In addition, as far as the vibrational eigenvalue calculation is concerned, as the authors write, VCI is more accurate than VPT2, which is very similar to the method they are employing. In fact, in previous publications the authors used to compare their VQDPT2 method against VCI to check its accuracy. In conclusion, the authors employed a less accurate method on a less accurate PES than the one already presented in the literature on specialist journals, claiming to have reached a better agreement with the experiments.

4) The authors present their results as nice spectroscopic profile of Lorentzian envelopes, typical of a spectroscopic manual. However, from their software they obtain a frequency value and the related intensity. In other words, they obtain a stick spectrum. I think it is not fair to report in the manuscript as well as in the SI a spectrum that looks like as it would be obtained from the Fourier transform of a spectroscopic signal without explaining that it is actually a stick spectrum, which is actually the case of experiments and molecular dynamics simulations for spectroscopic calculations. Even if their results look more eye-catching as Lorentzian envelope when compared with experiments, this is not what they actually calculated.

5) The authors claim the accuracy of their approach also by comparing with the experimental bending peak. However, the experimental bending band is not well reproduced: in the experiment it is sharp and intense, while in the simulation it is spread and with a low intensity. The authors cannot claim as they did that their intensities are accurate, while other methods have shown better accuracy in terms of intensity.

6) At pg.11, the authors claim that "It also signifies that, for the first time, accurate theoretical calculations for molecular systems of that size are possible." This is not true and the authors should report more accurately what has been published and presented in literature. For example, VPT2 have been applied to more complex systems and semiclassical methods, which have shown a quantum vibrational accuracy comparable to that of VCI and VPT2 methods, have been employed to perform comparable dimension water clusters spectra calculations on a CCSD(T)/VQZ PES.

Other issues:

1) In the abstract the authors write: "the first complete and accurate assignment of the IR spectra of the $H+(H_2O)_{21}$ cluster over the entire frequency region of the experiment." This is true only in part since the spectrum ranges from 700 to 3700 wavenumbers. In addition, all calculations performed in this MS are 89-dimensional, while the dimension of the cluster is 186. Thus, the authors cannot claim to have performed a full spectroscopic assignment.

2) In the Introduction the authors claim to have introduced a new protocol of new PES and anharmonic vibrational calculations. I do not agree with this overstatement, the protocol is not new and it is actually less accurate than previous ones, as explained above.

- 3) pg.9 "Our study demonstrates that the there": remove "the"
- 4) The authors claim their accuracy by virtue of the agreement of the free OH signal. However, this is not a good test because the mode is stiff and harmonic.
- 5) The authors write that "The site-specific IR spectrum reveals distinct structures for the internal ice-like and surface liquid-like four-coordinated water molecules that are the cornerstone of understanding the local structure of water in diverse environments." I think that this term of comparison, which is based exclusively on bonds and angles as shown in Table 1 is quite limited. What about the fundamental frequencies? Even at harmonic level these would have been enough for a comparison between the two phases.
- 6) at pg 12, the authors write: "The broad absorption around 2720 cm⁻¹ was rarely explored due to the weak intensity in the experimental spectrum. In addition, previous theoretical calculations have not been able to predict this spectral feature. The present calculation offers a clear-cut assignment of the 2720 cm⁻¹ band to the feature of proton defect, which involves the asymmetric O-H stretching, frustrated rotation and bending modes of H₃O⁺"
The authors should reconsider this statement which is what was already reported in the publication by Yu and Bowman. When referring to Fig.2 these authors consider that "Two PESs [WHBB and MB-pol] predict almost the same position and intensity of hydronium stretches at ~2700 cm⁻¹"
- 7) In the SI at pg 6, the authors show that it is not the full dimensional Hessian to be diagonalized but a reduced one. This procedure allow for easier normal mode assignment to each monomer, however, harmonic frequencies can differ up to 100 wavenumber respect to the full Hessian diagonalization. The author should report the difference between the block diagonal Hessian and the full Hessian eigenvalues. Is this procedure a biased starting point for the perturbative approach? The 89 modes considered in their VPT2 simulations should be biased by this choice. Why the authors still consider these "vibrational coordinates" comparable with the normal modes"? Can they prove that this approximation is not biasing their results?
- 8) Table S1 and S2 in the SI. For a better comparison, the authors should also add the experimental frequency values. Then MAE can be calculated with respect to these, instead of limiting their comparison by eye from the figure plots.
- 9) in the SI at pg. 6: "Gaussian16 progrom" It should be "program".
- 10) Eq S6: on the left 'PP' should not be in capital letters
- 11) Fig. S1: the label '4000' is cut and it looks like '400'

At the light of the considerations reported above, I think that the work presented is mainly of incremental type and more suitable for a specialistic journal, as it has been the case of other publications that I cited above. Finally, the authors should clean their draft from the numerous overstatements and report their results in the form of stick spectra.

Reviewer #2 (Remarks to the Author):

The protonated water clusters have been the subject of extensive experimental and theoretical studies over past decades, due to their great importance on understanding the fundamental role these simple ions play in aqueous chemistry. This manuscript provides a complete and clear assignment of the experimental spectroscopic features of protonated water cluster H⁺(H₂O)₂₁ based on high-level electronic wavefunction theory and anharmonic vibrational perturbation theory, and contributes to fully understanding the microscopic behavior of the excess proton in the water cluster at the atomic level. I thoroughly enjoyed the insights provided by this manuscript, and strongly believe it is worthy of publication, particularly as it extends and adds to the discussion of spectral signatures of ice-like and liquid-like water molecules. Overall, it is a very impressive achievement and I do believe that this paper will be of great interest to a broad range of the Nature Communications readers. Therefore, I recommend its publication in Nature Communications after a minor revision by considering the following comments.

- 1) The high-level CC calculations of the complex H⁺(H₂O)₂₁ cluster were carried out using the authors' previously developed EE-GMF fragmentation method. The authors may briefly address the accuracy and efficiency of this method in the main text, so that the readers could know the computational cost to achieve such high-level calculations. In addition, is the EE-GMF program open-source, and could anyone who is interested in this method use it for his/her own studies?
- 2) In the present work, the frustrated rotations of H₃O⁺ are all included in the resonance or

combination bands of the asym/sym OH stretch, umbrella and bending of H₃O⁺, but where is the fundamental of the frustrated rotation of H₃O⁺ in the IR spectra? As the readers may be concerned about these fundamental vibrations, the authors should make a clear description of all the fundamental vibrations of H₃O⁺, as well as their couplings, in the main text. In addition, the authors may also give the assignment of the vibrational peaks in the librational region.

3) The coupling between vibrational modes is essential for energy transfer. At present, the information on the vibrational coupling of the H-O-H bending mode of water is relatively lacking, even though the bending mode is an essential intermediate for the energy relaxation pathway. The novel energy relaxation pathway through bending overtone states found in this work provides very important information, hence the authors may discuss more on Fig. 4 to elucidate the vibrational energy transfer pathway of water.

4) In the comparison of the OH stretching features of different low-lying energy structures, there are a few unassigned signatures (denoted in black color) for isomers a₂, a₃ and a₄ in Fig. 5C. What makes these unassigned features different from features a₁? Could the authors provide more details on this?

Reviewer #3 (Remarks to the Author):

In this work, high level calculations of the vibrational absorption spectrum of the H⁺+(H₂O)₂₁ cluster are presented. In particular, an assignment of all the experimental absorption bands is proposed based on the calculations.

As far as I can tell, technically this work is of high quality. Combining highly accurate state of the art methods for both the electronic (fragment-based Coupled Cluster) and the nuclear (VQDPT2) problem, the frequencies of the experimental spectral bands are well reproduced. The agreement on relative intensities is less impressive (more comments on this would have been useful).

The manuscript is clearly written, and the band assignment is precisely described. My perplexity concerns the degree of broad interest of this work. By comparing the results presented here with those of the recent ref. 40 (Q. Yu, J. M. Bowman, J. Phys. Chem. A 124, 1167-1175 (2020)), I find essentially a general confirmation of the results presented there, with some improvements in the description of the bands at around 2200 cm⁻¹. As such, from the broad perspective of understanding these clusters, the present results look like refinements that do not substantially change the established picture. If this is not the case, then the important message of the work is lost amid the details of the band assignment description. In short, while the computational achievement is evident, I am not fully convinced the results themselves are of broad interest or of primary importance in the field. As such, I am not able to recommend publication in Nature Communications

6 July 2021

On behalf of all authors, I am submitting the revised manuscript titled “*The Complete Assignment of the Infrared Spectrum of the Protonated Water Cluster $H^+(H_2O)_{21}$* ” by Jinfeng Liu, Jinrong Yang, Xiao Cheng Zeng, Sotiris S. Xantheas, Kiyoshi Yagi, and Xiao He in which we address the comments of the reviewers. We thank the reviewers for taking the time to offer constructive comments to our manuscript.

In particular, the point-by-point response to the comments is as follows:

Reviewer #1 (Remarks to the Author):

This MS presents a tentative assignment of the $H^+(H_2O)_{21}$ water cluster IR spectrum. This system, and related ones of smaller dimension, has been the topic of several papers in the past, both computational and experimental ones, mainly published in specialized physical chemistry journals. This MS does not present any new experiment and it focuses on the assignment of a previous one. To reach this goal, the authors develop a PES and employ a method already present in the literature since 2008, which is called Second-order Vibrational Quasi-Degenerate Perturbation Theory (VQDPT2).

I report here below my specific considerations, before drawing my conclusions.

1) My main concern is about the methodological approach. In the past, perturbative approaches have been shown to break down. In a 2017 paper (J. Phys. Chem. A 2017, 121, 3056–3070) it has been reported about VPT2 (which is very similar to the present approach) applied to protonated water clusters and in trying to reproduce Ref.14 that “It can be difficult to decide which modes are described well in VPT2 and which ones are not.While the harmonic frequencies do not differ by more than ca. 100 cm^{-1} between different methods, the anharmonic ones are irreproducible and vary erratically between [VPT2 and others] methods....The huge discrepancy between methods and basis sets suggests that the anharmonic frequencies of these three [proton] modes are affected by the inherent instability of perturbation methods, and should be better replaced by their harmonic counterparts”. In another publication, the VPT2 methods have been shown to differ from more accurate approaches such as MCTDH and DMC by more than 500 wavenumbers in the Zundel cation, which is the smallest protonated water cluster. In a 2019 paper (J. Phys. Chem. A 2019, 123, 1399–1409), it has been reported that “The VPT2 method is comparatively cheaper for vibrational analysis, and it has been applied to the spectrum calculation of different sizes of protonated water cluster.10,11,31 Though many successes have been achieved using this method, it cannot fully deal with a very anharmonic system and thus cannot provide explanations to some important regions in spectra.1”.

The authors perform their calculations with VQDPT2, which for some aspects suffers from the same limitations of VPT2 (perturbative approach), and which has been introduced for the first time in 2008. The method is very briefly recalled in the SI, without details about its accuracy performance. In the literature, the accuracy of VQDPT2 has been compared with VCI on smaller protonated clusters but not with other methods. This would have benefited the broad readership of Nat. Commun. This would also have been highly desirable, given the citations that I reported in the previous paragraph. At the light of those considerations, the authors did not offer in their MS any method to compare with using the same PES to assess their results, at least at a reduced dimension. Instead, they prefer to accept that if the calculations are in some sort of agreement with the experiments, then they are accurate. I do not agree with this approach, since compensation of errors are

quite common in *ab initio* spectroscopy, where quite often harmonic results have shown to be more accurate than anharmonic ones on some systems.

Our response: We agree with the reviewer that VPT2 is not a suitable method for calculating the IR spectrum of protonated water clusters. One of the reasons of VPT2 failure is that VPT2 correction diverges in the presence of the vibrational resonance. The VPT2 equation is shown in page 6 of the supplementary material as

$$E_n^{\text{VPT2}} = \sum_{\mathbf{q} \neq \mathbf{n}} \frac{|\langle \Phi_n^{\text{HO}} | \delta V | \Phi_q^{\text{HO}} \rangle|^2}{E_n^{\text{HO}} - E_q^{\text{HO}}} \quad (\text{S13})$$

where Φ_n^{HO} is the harmonic oscillator wavefunction and

$$E_n^{\text{HO}} = \sum_{i=1}^f \left(n_i + \frac{1}{2} \right) \omega_i, \quad (\text{S14})$$

$$\delta V = \sum_{iik} c_{ijk} Q_i Q_j Q_k + \sum_{iijk} c_{iijk} Q_i^2 Q_j Q_k. \quad (\text{S15})$$

The occurrence of vibrational resonance means that there happens to be a vibrational state, \mathbf{q} , that is close in energy to a target state, \mathbf{n} , such that $E_n^{\text{HO}} \cong E_q^{\text{HO}}$. In such a case, the denominator of Eq. (S13) becomes zero, thereby leading to the divergence of the VPT2 energy. The protonated water clusters have many such resonance states, and thus VPT2 calculations result in an observation described in the 2017 paper (J. Phys. Chem. A 2017, 121, 3056–3070), “**the anharmonic ones are irreproducible and vary erratically**”.

Another deficiency of VPT2 comes from the usage of a 4th-order Taylor expansion or the quartic force field (QFF), Eq. (S15), for the PES. Note that VPT2 itself is not limited by any form of the PES, but that VPT2 with QFF is implemented in the Gaussian program and widely used to date. In general, QFF is a good choice for semi-rigid molecules; yet its applicability to soft molecules like protonated water clusters is questionable.

VQDPT2 has been developed to alleviate the divergence problem of perturbative approaches while keeping its cost-effectiveness since its beginning. The original paper in 2008 (PCCP 10, 1781 (2008)) showed that VQDPT2 was able to calculate the Fermi resonance states of small molecules (CO₂, H₂CO, and C₆H₆) as accurate as VCI. There has been a number of technical improvements since then. The most recent development uses local coordinates in place of standard normal coordinates in VQDPT2 calculation (Yagi and Sugita, submitted). We have also developed a multiresolution method to generate accurate anharmonic PES, which is a composite of QFF and grid PES (TCA 118, 681 (2007)). These methods have made feasible to compute the IR spectrum of H⁺(H₂O)₄ with high accuracy (JPCA 121, 2386 (2017)).

In response to the reviewer’s comment, we added the following lines in the main text on p. 5:

“VQDPT2 has been tested to be as accurate as VCI for small molecules (51, 52), but is scalable to many-mode systems. Recently, the method has been further improved by utilizing local coordinates and applied to strongly hydrogen bonded network in biomolecules (Yagi and Sugita, submitted). In this work, VQDPT2 has been carried out in 89 dimensions using coordinates localized to each molecule of the H⁺(H₂O)₂₁ cluster. (See the Materials and Methods section in the Supplementary Material for details).”

51. K. Yagi, S. Hirata, K. Hirao, *Phys. Chem. Chem. Phys.* **10**, 1781-1788 (2008).

52. K. Yagi, H. Otaki, *J. Chem. Phys.* **140**, 084113 (2014).

Details on the method are given in the supplementary material. The selection of quasi-degenerate states (P space) and non-degenerate states (Q space) based on N_{gen} and λ_{max} is described on page 5 of the SI:

“The P and Q space is constructed using two control parameters, N_{gen} and λ_{max} , for a target vibrational state of interest, \mathbf{n} . VSCF configurations that are quasi-degenerate to \mathbf{n} are searched in a configuration space $\{\mathbf{s}\}$ defined by λ_{max} as,

$$\lambda_{\mathbf{sn}} = \sum_{i=1}^f |s_i - n_i| \leq \lambda_{\text{max}} \quad (\text{S9})$$

The quasi-degenerate configurations found in the search are denoted $\{\mathbf{p}^{(1)}\}$. Then, the same search is carried out for each configuration of $\{\mathbf{p}^{(1)}\}$ to find the second generation of quasi-degenerate configurations, $\{\mathbf{p}^{(2)}\}$. The process is repeated N_{gen} times to obtain the P space configurations,

$$P = \{\mathbf{n}\} + \{\mathbf{p}^{(1)}\} + \{\mathbf{p}^{(2)}\} + \dots + \{\mathbf{p}^{(N_{\text{gen}})}\} = \{\mathbf{p}^m | m = 1, 2, \dots, N_p\} \quad (\text{S10})$$

The Q space is constructed by selecting configurations, \mathbf{q} , that satisfies the following condition,

$$\lambda_{\mathbf{q}\mathbf{p}^m} = \sum_{i=1}^f |q_i - p_i^m| \leq \lambda_{max} \quad (\text{S11})$$

Note that the Q space configurations are non-degenerate with any of the P space configurations. The energy differences in the denominator of Eq. (S7) are finite, and thus VQDPT2 is free of a divergence problem.”

Furthermore, sections of Relation of VQDPT2 with Other Vibrational Methods and Vibrational calculations based on local coordinates are described in page 6 – 8 of the supplementary material as the following.

“Relation of VQDPT2 with Other Vibrational Methods

Here, we briefly describe how VQDPT2 is related to other vibrational methods in the literature, namely, the second-order vibrational Møller Plesset (VMP2) (also known as the correlation corrected VSCF (cc-VSCF))^{16,17}, the second-order vibrational perturbation theory (VPT2)¹⁸, and the vibrational configuration interaction (VCI)¹⁹.

When the P space is reduced to one configuration, i.e., the target configuration (n), Eq. (S7) is rewritten as,

$$\left(H_{eff}^{(2)} \right)_{nm} = E_n^{VMP2} = \sum_{\mathbf{q} \neq n} \frac{|\langle \Phi_n^{VSCF} | \hat{H}_v | \Phi_q^{VSCF} \rangle|^2}{E_n^{(0)} - E_q^{(0)}}. \quad (\text{S12})$$

This equation is equivalent to VMP2. Therefore, VQDPT2 coincides with VMP2 if the target state has no quasi-generate states. Further simplification of Eq. (S12) gives VPT2 by replacing the VSCF solution with the harmonic solution, and by approximating the PES to a quartic force field (QFF),

$$E_n^{VPT2} = \sum_{\mathbf{q} \neq n} \frac{|\langle \Phi_n^{HO} | \delta V | \Phi_q^{HO} \rangle|^2}{E_n^{HO} - E_q^{HO}}, \quad (\text{S13})$$

where $|\Phi_n^{HO}\rangle$ is the harmonic oscillator wavefunction and

$$E_n^{HO} = \sum_{i=1}^f \left(n_i + \frac{1}{2} \right) \omega_i, \quad (\text{S14})$$

$$\delta V = \sum_{iik} c_{ijk} Q_i Q_j Q_k + \sum_{iik} c_{iik} Q_i^2 Q_j Q_k. \quad (\text{S15})$$

The integrals in Eq. (S13) can be carried out analytically, so that VPT2 is cost effective compared to other anharmonic vibrational methods. VPT2 is available in Gaussian16,²⁰ and has been widely used. However, the application of VPT2 to protonated water clusters is severely limited, because (1) the PES is highly anharmonic and beyond the applicability of QFF, and (2) the vibrational resonance caused by the motion of proton makes the perturbative expansion unstable [the energy difference in the denominator of Eq. (S13) is close to zero]. The drawbacks of VPT2 have been pointed out in the previous report.²¹

The VCI wavefunction is represented by a linear combination of VSCF configuration functions,

$$|\Psi_n^{VCI}\rangle = \sum_{\mathbf{m}} c_{n\mathbf{m}} \Phi_{\mathbf{m}}^{VSCF}. \quad (\text{S16})$$

The expansion coefficients are obtained by solving a secular equation,

$$\mathbf{H}\mathbf{c} = \mathbf{E}\mathbf{c}, \quad (\text{S17})$$

$$\mathbf{H}_{\mathbf{m}\mathbf{m}'} = \langle \Phi_{\mathbf{m}}^{VSCF} | \hat{H}_v | \Phi_{\mathbf{m}'}^{VSCF} \rangle. \quad (\text{S18})$$

VCI converges to the exact solution if all possible VSCF configurations are used in Eq. (S16). In practice, the expansion is truncated at some level to meet the computational cost. For example, the VCI space was constructed by restricting the level of excitation to 11, 10, 9, and 8 for one-, two-, three-, and four-mode excitations, respectively, in VCI calculations of a H_9O_4^+ moiety in $\text{H}^+(\text{H}_2\text{O})_{21}$.^{21,22} The VCI dimension was about 140,000.

The first-order VQDPT in Eq. (S6) is equivalent to VCI truncated to the P space, and VQDPT2 is an improvement over the truncated VCI. In the original article,^{14,15} the benchmark calculations have shown that VQDPT2 calculations with $N_{gen}=3$ and $\lambda_{max}=4$ were comparable to large VCI calculations. In the present work, the number of P and Q space configurations was about 900 and 300 million, respectively, in the VQDPT2 calculations of $\text{H}^+(\text{H}_2\text{O})_{21}$.

Vibrational calculations based on local coordinates

Although conventional vibrational calculations have been carried out in terms of normal coordinates, local coordinates have gained more attention in recent years. We have developed optimized coordinate VSCF (oc-VSCF) method²³, which yields a set of variationally optimal coordinates to describe the system. Application to H_9O_4^+ has shown that coordinates localized to each molecule (H_3O^+ and H_2O) is better than delocalized, normal coordinates.²⁴ The result is consistent with Wang and Bowman,²⁵ who proposed to employ coordinates localized to each molecule for VCI calculations of molecular clusters.

Very recently, we have developed VQDPT2 calculations based on local coordinates.²⁶ The coordinates are obtained by diagonalizing the Hessian matrix in a block of user-specified group of atoms. The PES was represented as a sum of the intra-group anharmonic PES (V_g) and inter-group harmonic coupling ($c_{gg'}$) as,

$$V \approx \sum_g V_g(\mathbf{Q}_g) + \sum_{g>g'} c_{gg'} \mathbf{Q}_g \mathbf{Q}_{g'} \quad (\text{S19})$$

where g is an index of the groups. The intra-group PES is generated in a conventional way, whereas the inter-group coupling is truncated at the harmonic level. Because the inter-group coupling is obtained from the Hessian matrix, the cost of PES generation is drastically reduced. The validity of the approximation was tested through an application to a strong hydrogen bond network in biomolecules.^{26''}

(14) Yagi, K.; Hirata, S.; Hirao, K. Vibrational quasi-degenerate perturbation theory: applications to fermi resonance in CO_2 , H_2CO , and C_6H_6 . *Phys. Chem. Chem. Phys.* **2008**, *10*, 1781-1788.

(15) Yagi, K.; Otaki, H. Vibrational quasi-degenerate perturbation theory with optimized coordinates: Applications to ethylene and trans-1,3-butadiene. *J. Chem. Phys.* **2014**, *140*, 084113.

(16) Christiansen, O. Moller-Plesset perturbation theory for vibrational wave functions. *J. Chem. Phys.* **2003**, *119*, 5773-5781.

(17) Norris, L. S.; Ratner, M. A.; Roitberg, A. E.; Gerber, R. B. Moller-Plesset perturbation theory applied to vibrational problems. *J. Chem. Phys.* **1996**, *105*, 11261-11267.

(18) Barone, V. Anharmonic vibrational properties by a fully automated second-order perturbative approach. *J. Chem. Phys.* **2005**, *122*, 014108.

(19) Christoffel, K. M.; Bowman, J. M. Investigations of Self-Consistent Field, SCF CI and Virtual State Configuration-Interaction Vibrational Energies for a Model 3-mode System. *Chem. Phys. Lett.* **1982**, *85*, 220-224.

(20) Frisch, M. J.; Trucks, G. W.; Schlegel, H. B.; Scuseria, G. E.; Robb, M. A.; Cheeseman, J. R.; Scalmani, G.; Barone, V.; Petersson, G. A.; Nakatsuji, H.; Li, X.; Caricato, M.; Marenich, A. V.; Bloino, J.; Janesko, B. G.; Gomperts, R.; Mennucci, B.; Hratchian, H. P.; Ortiz, J. V.; Izmaylov, A. F.; Sonnenberg, J. L.; Williams-Young, D.; Ding, F.; Lipparini, F.; Egidi, F.; Goings, J.; Peng, B.; Petrone, A.; Henderson, T.; Ranasinghe, D.; Zakrzewski, V. G.; Gao, J.; Rega, N.; Zheng, G.; Liang, W.; Hada, M.; Ehara, M.; Toyota, K.; Fukuda, R.; Hasegawa, J.; Ishida, M.; Nakajima, T.; Honda, Y.; Kitao, O.; Nakai, H.; Vreven, T.; Throssell, K.; Montgomery, J. A., Jr.; Peralta, J. E.; Ogliaro, F.; Bearpark, M. J.; Heyd, J. J.; Brothers, E. N.; Kudin, K. N.; Staroverov, V. N.; Keith, T. A.; Kobayashi, R.; Normand, J.; Raghavachari, K.; Rendell, A. P.; Burant, J. C.; Iyengar, S. S.; Tomasi, J.; Cossi, M.; Millam, J. M.; Klene, M.; Adamo, C.; Cammi, R.; Ochterski, J. W.; Martin, R. L.; Morokuma, K.; Farkas, O.; Foresman, J. B.; Fox, D. J. Gaussian 16, Revision C.01, Gaussian, Inc., Wallingford CT. **2016**.

(21) Wang, H.; Agmon, N. Reinvestigation of the Infrared Spectrum of the Gas-Phase Protonated Water Tetramer. *J. Phys. Chem. A* **2017**, *121*, 3056-3070.

(22) Yu, Q.; Bowman, J. M. Tracking Hydronium/Water Stretches in Magic $\text{H}_3\text{O}^+(\text{H}_2\text{O})_{20}$ Clusters through High-level Quantum VSCF/VCI Calculations. *J. Phys. Chem. A* **2020**, *124*, 1167-1175.

(23) Yagi, K.; Keceli, M.; Hirata, S. Optimized coordinates for anharmonic vibrational structure theories. *J. Chem. Phys.* **2012**, *137*, 204118.

(24) Yagi, K.; Thomsen, B. Infrared Spectra of Protonated Water Clusters, $\text{H}^+(\text{H}_2\text{O})_4$, in Eigen and Zundel Forms Studied by Vibrational Quasi-Degenerate Perturbation Theory. *J. Phys. Chem. A* **2017**, *121*, 2386-2398.

(25) Wang, Y. M.; Bowman, J. M. Ab initio potential and dipole moment surfaces for water. II. Local-monomer calculations of the infrared spectra of water clusters. *J. Chem. Phys.* **2011**, *134*, 154510.

(26) Yagi, K.; Sugita, Y. Anharmonic Vibrational Calculations Based on Group Localized Coordinates: Applications to Internal Water Molecules in Bacteriorhodopsin. **2021**, submitted.

In addition, I cannot see enough novelty in their work since the VPT2 protonated water spectrum has been already published in 2011 by one of the authors. More recently, VCI has been employed on a more accurate PES (vide infra). For these reasons, I consider this work mainly of incremental type and with a reduced novelty in the field.

Our response: As mentioned above, VQDPT2 is far superior to VPT2. By the way, the VPT2 work was reported by Torrent-Sucarrat and Anglada, JCTC 2011, and none of us co-authored that paper. The comparison of the present work with VCI work by Yu and Bowman (JPCA 2021) is discussed in the reply to comment (3). Besides the technicalities, the new insights into the $H^+(H_2O)_{21}$ cluster obtained in this study are (1) the assignment of IR bands of the proton defect in a range of $1700 - 2800 \text{ cm}^{-1}$, (2) four-coordinated water molecules on the surface (AADD^s) are liquid-like, whereas the one in the interior (AADDⁱ) is ice-like, (3) the water-water interaction invokes a relaxation of O-H stretching vibrational energy of water molecules mediated by bending overtone of nearby water molecules. These points are mentioned clearly in the abstract and the discussion section. In the result section, we organized subsections, “Band assignment”, “Two types of AADD water”, and “Intermolecular couplings of water molecules”. We hope these changes have made clear the novelty of the present work.

2) *The methodological approach strongly depends on the perturbative application point. For these reasons, when a perturbative approach is adopted for spectroscopic calculations, an alternative approach based on MD is employed to check how relevant other geometry minima are in the experimental spectroscopic signal. This may include ab initio molecular dynamics, path integral molecular dynamics and semiclassical molecular dynamics. The authors try to partial remedy to this limitation by looking at 4 other similar minima. Clearly these calculations cannot be considered representative of the thousands of possible minima. Actually, the experiments have provided only indirect evidence that the protonated water is the one at the top of the cluster. It would have been much more interesting if the authors had shown that by means of a dynamical approach that indeed would have shown that the predominant spectroscopic signal is originated from these configurations. Instead, the authors enforced these few configurations into their perturbative approach.*

Our response: We thank the reviewer for making this comment. The *ab initio* molecular dynamics (AIMD) simulation of $H^+(H_2O)_{21}$ cluster at the coupled cluster level is very time consuming, and it would become even worse when the quantum effects are considered by using the path integral molecular dynamics. Hence, we did not use these procedures in the present work. Some related works are underway in our lab, in which we can carry out *ab initio* path integral molecular dynamics simulation of $H^+(H_2O)_{21}$ cluster by using a high dimensional machine learning potential. On the other hand, Agmon and co-workers have performed a lot of AIMD simulations of protonated water clusters of different sizes, and calculated their anharmonic spectra, which make more understanding of the protonated water structures.

We have added some introduction and cited Agmon’s and co-workers’ corresponding papers as follows:

“Agmon and co-workers have studied the IR spectra of protonated water clusters of different sizes by using *ab initio* molecular dynamics simulations (30-33), which contributed more understanding of the protonated water structures.”

30. W. Kulig, N. Agmon, J. Phys. Chem. B 118, 278-286 (2014).

31. W. Kulig, N. Agmon, Nat. Chem. 5, 29-35 (2013).

32. W. Kulig, N. Agmon, Phys. Chem. Chem. Phys. 16, 4933-4941 (2014).

33. H. Wang, N. Agmon, J. Phys. Chem. A 121, 3056-3070 (2017).

3) *At pg. 4, the authors write: “Yu and Bowman(40) have shown that the higher level, vibrational configuration interaction (VCI) method achieves a significant improvement over VPT2. Nevertheless, in their VCI calculation, the potential energy surface (PES) of $H^+(H_2O)_{21}$ (41) was represented as a sum of potential energy functions for small clusters, $H^+(H_2O)_n$ ($n = 1 - 4$) and $(H_2O)_n$, ($n = 1 - 3$), without directly accounting for the electronic structure of the full cluster. Therefore, a more sophisticated quantum treatment of both the electronic and vibrational structure of the $H^+(H_2O)_{21}$ cluster is required to produce computed spectra to accurately assign the experimental spectral features.” I do agree that VCI is more accurate than VPT2 but I do not agree that the previous PES and method (VCI) is less accurate than the one employed in the MS. Present calculations are at the level of CCD/aug-cc-pVDZ theory, while previous PES is at the level of CCSD(T)-F12/aVQZ (see J. Phys. Chem. A 2019, 123, 1399–1409), which is by far more accurate both in the ab initio method (CCSD(T) vs CCD) and basis set (aVQZ vs aug-cc-pVDZ). Also, the previous PES includes 2 and 3 body interactions (and 4 body interactions are calculated at the level of MP2/aVTZ) for all degrees of freedom and the approach has proved to be basically equivalent to the full-dimensional (full-body) PES.*

Conversely, the PES of this work, which is the only methodological novelty of the MS, accounts for 1 and 2 body interactions only for the chosen degrees of freedom (i.e. 89 out of 186), while the higher order interactions are considered as a standard typical MD force field. In addition, as far as the vibrational eigenvalue calculation is concerned, as the authors write, VCI is more accurate than VPT2, which is very similar to the method they are employing. In fact, in previous publications the authors used to compare their VQDPT2 method against VCI to check its accuracy. In conclusion, the authors employed a less accurate method on a less accurate PES than the one already presented in the literature on specialist journals, claiming to have reached a better agreement with the experiments.

Our response: First of all, we agree with the reviewer that the wording was not appropriate in the original manuscript regarding, “Therefore, a more sophisticated quantum treatment of both the electronic and vibrational structure of the $H^+(H_2O)_{21}$ cluster is required to produce computed spectra to accurately assign the experimental spectral features”. This line is removed in the revised manuscript.

The level of theory in this work and the previous work by Yu and Bowman is not directly comparable. The present method explicitly treats the electrons and vibrations of the $H^+(H_2O)_{21}$ cluster as a whole, while Yu and Bowman performed explicit calculations only for fragments of the $H^+(H_2O)_{21}$ cluster. Thus, we may claim that our approach is more accurate for treating the whole system; however, each fragment of Yu and Bowman is treated with higher level of theory than ours. We explain more specifically below.

As for the PES, the previous potential energy function (PEF) was fitted at the level of CCSD(T)-F12/aVQZ (J. Phys. Chem. A 2019, 123, 1399–1409) for the hydronium H_3O^+ monomer. For the multi-body interactions between H_3O^+ and H_2O , as well as H_2O and H_2O , they were fitted at diverse *ab initio* levels of CCSD(T)/aVTZ, CCSD(T)/aVDZ, and MP2/aVTZ. Thus, the *ab initio* level used to fit their PEFs are higher than that the one used in our present work. However, the PESs of the $H^+(H_2O)_{21}$ cluster were constructed by simple summation of H_3O^+ and H_2O monomers and multimers (dimer, trimer, and tetramer) without considering the environmental polarization effects.

On the other hand, in the present work with the EE-GMF method, all monomers and dimers were explicitly treated by the standard *ab initio* calculations with the electrostatically embedding scheme. The scheme accounts for the environmental effect, and the effect of polarizability and hydrogen-bond cooperativity. In the dimer calculations, one- and two-body kinetic, Coulomb, exchange, and correlation interactions are included nearly exactly at the CCD level, and the three- and all higher-order Coulomb effects are also implicitly included through the electrostatic embedding. It is this embedding approach that makes the Bethe-Goldstone expansion rapidly convergent and the truncation after the dimer terms sufficiently accurate. Hence, the EE-GMF method is capable of reproducing the *ab initio* properties of molecular clusters with negligible errors, as can be seen from our previous papers (J. Chem. Theory Comput. 2017, 13, 2021; Chem. Sci. 2018, 9, 2065; Phys. Chem. Chem. Phys. 2020, 22, 12341).

As for the vibrational calculations, the previous VCI calculation was performed for a fragment of the cluster independently, i.e., a $H_3O^+(H_2O)_3$ moiety in 15 dimension and each water molecules in 3 dimension. The VCI calculation incorporated sufficient number of VSCF excited states (~140,000 for $H_3O^+(H_2O)_3$), and thus yields accurate, reliable results in the selected degrees of freedom. In this treatment, however, the vibrational interaction of $H_3O^+(H_2O)_3$ and other water is missing. Furthermore, VCI of each water molecules treated two O-H stretching and one H-O-H bending modes neglecting the librational modes.

On the other hand, the VQDPT2 calculation in this work was performed in 89 dimensions (9 dimensions of H_3O^+ and 4 dimensions of each H_2O) including the intermolecular interaction explicitly. The size of the largest *P* space (quasi-degenerate states) treated by VCI was 920, and the complementary *Q* space was 333 million.

We have accordingly modified the main text on page 4 as follows:

“Nevertheless, the calculation was based on a potential energy surface (PES) of $H^+(H_2O)_{21}$ represented as a sum of potential energy functions (PEFs) of small clusters, $H^+(H_2O)_n$ ($n = 1 - 4$) (45-47) and $(H_2O)_n$ ($n = 1 - 3$) (48), derived from *ab initio* electronic structure calculations. VCI calculations were also carried out for fragments of the cluster, $H_3O^+(H_2O)_3$ (15 dimension) and each H_2O (3 dimension). Theoretical calculations that account for the electronic and vibrational structure of the full $H^+(H_2O)_{21}$ cluster remain a challenge.”

45. Q. Yu, J. M. Bowman, J. Am. Chem. Soc. **139**, 10984-10987 (2017).

46. C. Qu, Q. Yu, J. M. Bowman, Annu. Rev. Phys. Chem. **69**, 151-175 (2018).

47. J. P. Heindel, Q. Yu, J. M. Bowman, S. S. Xantheas, J. Chem. Theory Comput. **14**, 4553-4566 (2018).

48. Y. M. Wang, X. C. Huang, B. C. Shepler, B. J. Braams, J. M. Bowman, *J. Chem. Phys.* **134**, 094509 (2011).

We also performed CCSD/aug-cc-pVTZ calculations to validate the accuracy of our calculations as stated on page 6 of the revised manuscript:

“Note that the correction of higher-level electronic correlation effects by employing CCSD/aug-cc-pVTZ calculations supports our present predictions (see Figure S3 of the Supplementary Material), which substantiates the effectiveness of the CCD/aug-cc-pVDZ level in computing and interpreting the IR spectral features.”

We added the VCI spectrum in Fig. 2 and the following lines in the main text on page 6:

“The previous VCI spectrum matches well with the present result. Some notable differences are: (1) VCI gives no signal in a range of 600 – 900 cm^{-1} because the librational modes of H_2O were excluded from the calculation. (2) The IR band shape in a range of 1700 – 2800 cm^{-1} appears different, where VCI exhibits a strong, broad band around 1950 cm^{-1} and diminishes beyond 2200 cm^{-1} , whereas VQDPT2 yields sharp peaks up to 2700 cm^{-1} . Nevertheless, the overall agreement of the IR spectra obtained by two different theoretical approaches indicates the robustness of the calculated results.”

In the Discussion section, we added a paragraph on page 12:

“The present calculation is complementary with the previous VCI calculation based on *ab initio* PEF by Yu and Bowman. On one hand, the PEFs of $\text{H}^+(\text{H}_2\text{O})_n$ ($n = 1 - 4$) and $(\text{H}_2\text{O})_n$, ($n = 1 - 3$) were derived at the mixed high electronic structure levels of CC and MP2, but they were simply summed to construct the PEF of $\text{H}^+(\text{H}_2\text{O})_{21}$, whereas the PES in our calculation is computed for $\text{H}^+(\text{H}_2\text{O})_{21}$ by the fragment-based CC in an electrostatically embedding scheme at the level of CCD/aug-cc-pVDZ. On the other hand, VCI was carried for a $\text{H}_3\text{O}^+(\text{H}_2\text{O})_3$ moiety in 15 dimension incorporating $\sim 140,000$ of VCI states and other water molecules in 3 dimension, whereas VQDPT2 was performed for $\text{H}^+(\text{H}_2\text{O})_{21}$ in 89 dimension incorporating $\sim 1,000$ of quasi-degenerate states by VCI and hundreds of millions of non-degenerate states by perturbation. There are multiple measures on the level accuracy, and the two approaches are complementary with each other. Nonetheless, the resulting IR spectra exhibit an overall agreement, which substantiates the robustness of the theory even for such a complex system as $\text{H}^+(\text{H}_2\text{O})_{21}$.”

We added following discussion in the Supplementary Material:

“With the EE-GMF fragmentation method, all of the monomers and dimers were explicitly treated through the standard *ab initio* calculations with the electrostatically embedding scheme to account for the environmental effect, which ensures the effect of polarizability and hydrogen-bond cooperativity to be taken into account. Through the dimer calculations, one- and two-body kinetic, Coulomb, exchange, and correlation interactions are included nearly exactly at the CCD level. Through the embedding, three- and all higher-order Coulomb effects are also implicitly included. It is this embedding approach that makes the Bethe-Goldstone expansion rapidly convergent and the truncation after the dimer terms sufficiently accurate.”

4) The authors present their results as nice spectroscopic profile of Lorentzian envelopes, typical of a spectroscopic manual. However, from their software they obtain a frequency value and the related intensity. In other words, they obtain a stick spectrum. I think it is not fair to report in the manuscript as well as in the SI a spectrum that looks like as it would be obtained from the Fourier transform of a spectroscopic signal without explaining that it is actually a stick spectrum, which is actually the case of experiments and molecular dynamics simulations for spectroscopic calculations. Even if their results look more eye-catching as Lorentzian envelope when compared with experiments, this is not what they actually calculated.

Our response: We have indicated that in the caption of Figure 2:

“The harmonic and VQDPT2 spectra are broadened using Lorentz functions with the full-width at half-maximum (FWHM) of 5 cm^{-1} . The raw stick spectrum is shown in Figure S1 of the Supplementary Material.”

Figure S1. IR stick (black drop lines) and Lorentz broadened (red line) spectra of the $\text{H}^+(\text{H}_2\text{O})_{21}$ cluster at the Eigen state computed through VQDPT2 method at the fragment-based CCD/aug-cc-pVDZ level.

5) The authors claim the accuracy of their approach also by comparing with the experimental bending peak. However, the experimental bending band is not well reproduced: in the experiment it is sharp and intense, while in the simulation it is spread and with a low intensity. The authors cannot claim as they did that their intensities are accurate, while other methods have shown better accuracy in terms of intensity.

Our response: In the previous theoretical studies on $\text{H}^+(\text{H}_2\text{O})_{21}$ shown in Figure 2, the calculated bending band was also predicted to be with a low intensity. Our current study agrees with previous simulated results. The bending band is usually weak in other clusters (see the following figure from Science 308, 1765 (2005)), so the sharp, intense peak of bending mode seems unique in $\text{H}^+(\text{H}_2\text{O})_{21}$.

Editorial Note: From Headrick, J.M. et al. Spectral signatures of hydrated proton vibrations in water clusters. 541 *Science* **308**, 1765-1769 (2005). Reprinted with permission from AAAS.

6) At pg.11, the authors claim that “It also signifies that, for the first time, accurate theoretical calculations for molecular systems of that size are possible.” This is not true and the authors should report more accurately what has been published and presented in literature. For example, VPT2 have been applied to more complex

systems and semiclassical methods, which have shown a quantum vibrational accuracy comparable to that of VCI and VPT2 methods, have been employed to perform comparable dimension water clusters spectra calculations on a CCSD(T)/VQZ PES.

Our response: We thank the reviewer for making this comment. We have deleted this line.

Other issues:

1) *In the abstract the authors write: “the first complete and accurate assignment of the IR spectra of the H⁺(H₂O)₂₁ cluster over the entire frequency region of the experiment.” This is true only in part since the spectrum ranges from 700 to 3700 wavenumbers. In addition, all calculations performed in this MS are 89-dimensional, while the dimension of the cluster is 186. Thus, the authors cannot claim to have performed a full spectroscopic assignment.*

Our response: We thank the reviewer for making this comment. We changed the title of the manuscript to “Towards Complete Assignment of the Infrared Spectrum of the Protonated Water Cluster H⁺(H₂O)₂₁”

We also changed the sentence in the Abstract as follows:

“We present a new protocol for the calculation of the infrared (IR) spectra of complex systems, which combines fragment-based Coupled Cluster and anharmonic vibrational quasi-degenerate perturbation theory and demonstrate its accuracy towards the complete and accurate assignment of the IR spectra of the H⁺(H₂O)₂₁ cluster”.

2) *In the Introduction the authors claim to have introduced a new protocol of new PES and anharmonic vibrational calculations. I do not agree with this overstatement, the protocol is not new and it is actually less accurate than previous ones, as explained above.*

Our response: We have made an explanation in the reply to the above comment (3).

3) pg.9 “Our study demonstrates that the there”: remove “the”

Our response: We have removed “the” from that sentence.

4) *The authors claim their accuracy by virtue of the agreement of the free OH signal. However, this is not a good test because the mode is stiff and harmonic.*

Our response: We agree with the reviewer and this line is now removed. Besides, the free OH stretch is not “stiff and harmonic”. The harmonic frequency was obtained around 3890 cm⁻¹ on average, and the anharmonic correction was 200 cm⁻¹. The free-OH of the harmonic spectrum was out of the range of Figure 2 in the original manuscript. It is now extended to 3900 cm⁻¹ to make it visible.

5) *The authors write that “The site-specific IR spectrum reveals distinct structures for the internal ice-like and surface liquid-like four-coordinated water molecules that are the cornerstone of understanding the local structure of water in diverse environments.” I think that this term of comparison, which is based exclusively on bonds and angles as shown in Table 1 is quite limited. What about the fundamental frequencies? Even at harmonic level these would have been enough for a comparison between the two phases.*

Our response: We have added more discussion in the “Two types of AADD water” section of the revised manuscript as follows:

“The difference of the two types of AADD water molecule in the H⁺(H₂O)₂₁ cluster is also addressed in comparison with the experimentally observed bulk spectra of ice Ih (57) and liquid water (58), as shown in Figure S7 of the Supplementary Material. The O-H stretching frequency of liquid water shows blue shift by ~180 cm⁻¹ with reference to ice Ih, which is in good agreement with the present calculation (shifted by ~165-185 cm⁻¹) and further proves the existence of two types of AADD water molecule in the H⁺(H₂O)₂₁ cluster.”

57. J. E. Bertie, E. Whalley, *J. Chem. Phys.* **40**, 1637 (1964).

58. J. E. Bertie, Z. D. Lan, *Appl. Spectrosc.* **50**, 1047-1057 (1996).

Figure S7. Comparison of the experimental bulk IR spectrum of ice Ih and liquid water in the O-H stretching region. These data are extracted from refs. 57-58.

6) at pg 12, the authors write: “The broad absorption around 2720 cm^{-1} was rarely explored due to the weak intensity in the experimental spectrum. In addition, previous theoretical calculations have not been able to predict this spectral feature. The present calculation offers a clear-cut assignment of the 2720 cm^{-1} band to the feature of proton defect, which involves the asymmetric O-H stretching, frustrated rotation and bending modes of H_3O^+ ” The authors should reconsider this statement which is what was already reported in the publication by Yu and Bowman. When referring to Fig.2 these authors consider that “Two PESs [WHBB and MB-pol] predict almost the same position and intensity of hydronium stretches at $\sim 2700 \text{cm}^{-1}$ ”

Our response: We thank the reviewer for making this comment. In Yu and Bowman’s study, they indicated that the feature around $\sim 2700 \text{cm}^{-1}$ was due to the hydronium stretches, but they did not provide detailed information of the hydronium’s motions. In the present work, we give a clear and more detailed assignment for this proton defect signature, which is a combination band involving the asymmetric O-H stretching, frustrated rotation and bending modes of H_3O^+ .

We have modified our statement on page 7 of the revised manuscript as follows:

“The broad absorption around 2720 cm^{-1} was rarely explored due to the weak intensity in the experimental spectrum. The present calculation offers a clear-cut assignment of the 2720 cm^{-1} band to a resonance state of the asymmetric O-H stretching and a combination tone of the frustrated rotation and H_3O^+ bend

$(V_{\text{H}_3\text{O}^+}^{a^1, r+b}, V_{\text{H}_3\text{O}^+}^{a^2, r+b})$.”

7) In the SI at pg 6, the authors show that it is not the full dimensional Hessian to be diagonalized but a reduced one. This procedure allow for easier normal mode assignment to each monomer, however, harmonic frequencies can differ up to 100 wavenumber respect to the full Hessian diagonalization. The author should report the difference between the block diagonal Hessian and the full Hessian eigenvalues. Is this procedure a biased starting point for the perturbative approach? The 89 modes considered in their VPT2 simulations should be biased by this choice. Why the authors still consider these “vibrational coordinates” comparable with the normal modes”? Can they prove that this approximation is not biasing their results?

Our response: The differences in the frequencies are given in Table S4 of the supplementary material. The harmonic coupling terms are all included in the vibrational Hamiltonian in VQDPT2 calculations, and in that sense, there is no bias. Local coordinates are known to be better for molecular clusters, and commonly used nowadays. In the previous study on $\text{H}^+(\text{H}_2\text{O})_4$, we have found that variational optimized coordinates are localized to each molecule. The VCI study by Yu and Bowman also use the same type of coordinates.

8) Table S1 and S2 in the SI. For a better comparison, the authors should also add the experimental frequency values. Then MAE can be calculated with respect to these, instead of limiting their comparison by eye from the figure plots.

Our response: We thank the reviewer for making this comment. We have added the experimental frequency values in Tables S1-S3 of the Supplementary Material as follows:

Table S1. The assignments of the bands associated with H_3O^+ . Notations: frustrated rotation (rot), umbrella vibration (umb), H-O-H bending (bend), libration (lib), sym (symmetric O-H stretching), asym (asymmetric O-H stretching), overtone (Ovtn), combination band (Comb).

Exp./ cm^{-1}	VQDPT2/ cm^{-1}	Weight	Intensity	Mode
940	943	0.201	158	H_3O^+ rot
*	951	0.297	21	H_3O^+ rot
*	983	0.731	5	H_3O^+ rot
1220	1267	0.930	307	H_3O^+ umb
*	1728	0.101	4	H_3O^+ bend
*	1733	0.197	1	H_3O^+ bend
1746	1791	0.035	74	H_3O^+ asym ¹
*	1801	0.102	63	Comb(H_3O^+ rot + H_2O lib)
*	1812	0.366	60	Comb(H_3O^+ rot + H_2O lib)
*	1839	0.088	53	Comb(H_3O^+ rot + H_2O lib)
*	1918	0.085	51	Comb(H_3O^+ rot + H_2O lib)
1949	1949	0.051/asym ¹ 0.032/asym ² 0.140/Comb	160	H_3O^+ asym ¹ , H_3O^+ asym ² , Comb(H_3O^+ umb + H_3O^+ rot + H_2O lib)
*	1954	0.278	77	Comb(H_3O^+ umb + H_2O lib)
*	1974	0.166	57	Comb(H_3O^+ umb + H_2O lib)
*	2023	0.382	81	Comb(H_3O^+ umb + H_2O lib)
2015	2035	0.113	267	H_3O^+ asym ²
*	2052	0.054/asym ¹ 0.195/Comb	185	H_3O^+ asym ¹ , Comb(H_3O^+ umb + H_2O lib)
*	2068	0.026/asym ¹ 0.429/Comb	95	H_3O^+ asym ¹ , Comb(H_3O^+ umb + H_2O lib)
*	2138	0.024/asym ² 0.261/Comb	91	H_3O^+ asym ² , Comb(H_3O^+ umb + H_2O lib)
2220	2261	0.221/asym ² 0.360/Comb	529	H_3O^+ asym ² , Comb(H_3O^+ umb + H_3O^+ rot)
*	2286	0.172/asym ¹ 0.333/Comb	406	H_3O^+ asym ¹ , Comb(H_3O^+ umb + H_3O^+ rot)
*	2308	0.037/asym ¹ 0.485/Comb	90	H_3O^+ asym ¹ , Comb(H_3O^+ rot + H_2O lib)
2400	2409	0.296	194	H_3O^+ sym
*	2417	0.132/sym 0.302/Comb	83	H_3O^+ sym, Comb(H_3O^+ bend + H_2O lib)
*	2420	0.066/sym 0.195/Comb	62	H_3O^+ sym, Comb(H_3O^+ bend + H_2O lib)

*	2438	0.025/asym ² 0.432/Comb	70	H ₃ O ⁺ asym ² , Comb(H ₃ O ⁺ bend + H ₂ O lib)
*	2513	0.739	59	Ovtn (H ₃ O ⁺ umb)
2720	2714	0.040/asym ² 0.612/Comb	115	H ₃ O ⁺ asym ² , Comb(H ₃ O ⁺ bend + H ₃ O ⁺ rot)
*	2726	0.061/asym ¹ 0.586/Comb	168	H ₃ O ⁺ asym ¹ , Comb(H ₃ O ⁺ bend + H ₃ O ⁺ rot)
MAE ^x	24.0			

^x Mean absolute error.

Table S2. The assignments of the librational modes of neutral water.

Type	Exp./cm ⁻¹	VQDPT2/cm ⁻¹	Weight	Intensity
DDA^h	*	696	0.348	91
AAD	*	707	0.397	52
AAD	*	714	0.235	14
DDA^h	*	720	0.367	74
AAD	*	725	0.288	23
DDA^h	*	746	0.459	91
AADD^s	*	777	0.409	82
AAD	*	795	0.554	81
AAD	~840	855	0.248	293
AAD	*	862	0.379	63
AADD^s	*	893	0.209	30
AAD	*	908	0.265	190
AAD	*	923	0.319	39
AAD	*	963	0.487	49

Table S3. The assignments of the bands (**a-i**, in Figure 3) associated with the OH stretching motions of the neutral water molecules. Notations: overtone (Ovtn), combination band (Comb).

Type	Mode	HB partner ^x	Exp./cm ⁻¹	VQDPT2/cm ⁻¹	Weight	IR intensity	Band
AAD	Ovtn (bends)	*	3110	3166	0.352	188	i
AAD+DDA^d	Comb (bends)	*	3206	3269	0.121	29	h
AAD	OH	DDA ^d	3230	3284	0.368	332	g
AADDⁱ	asym	AADD ^s , AADD ^s	3304	3304	0.526	514	f
AAD	OH	DDA ^d	*	3323	0.860	646	e
AAD	OH	DDA ^d	3335	3327	0.660	317	
AAD	OH	AADD ^s	*	3361	0.449	545	
DDA^h	sym	AADD ^s , AAD	*	3378	0.106	214	d
AAD	OH	AADD ^s	*	3383	0.345	636	
DDA^h	asym	AADD ^s , AAD	3390	3388	0.126	636	

AAD	OH	AADD ^s	*	3392	0.157	226	
AADD ^s	asym	AAD, AAD	*	3402	0.363	732	
AADD ^s	sym	AAD, AAD	*	3411	0.145	278	
AAD	OH	AAD	*	3429	0.711	425	c
AAD	OH	AAD	*	3438	0.818	431	
DDA ^h	sym	AADD ^s , AAD	*	3444	0.377	407	
AAD	OH	AAD	3450	3450	0.568	697	
DDA ^h	asym	AADD ^s , AAD	*	3452	0.189	229	
DDA ^h	sym	AADD ^s , AAD	*	3453	0.188	281	
AADD ^s	sym	AAD, AAD	*	3459	0.442	254	
AADD ^s	OH	AAD	3525	3472	0.668	451	
AADD ^s	asym	AAD, AAD	*	3476	0.383	305	
AADD ^s	OH	AAD	*	3494	0.772	277	
DDA ^d	sym	AAD, AAD	*	3527	0.913	174	
DDA ^d	asym	AAD, AAD	3575	3529	0.998	506	a
DDA ^d	sym	AAD, AAD	*	3538	0.917	106	
DDA ^d	sym	AAD, AAD	*	3539	0.887	221	
DDA ^d	asym	AAD, AAD	*	3547	0.977	452	
DDA ^d	asym	AAD, AAD	3595	3550	0.976	564	
MAE ^y				32.7			

^xThe type of hydrogen-bonded water molecules directly involved in each vibrational mode.

^yMean absolute error.

9) in the SI at pg. 6: "Gaussian16 program" It should be "program".

Our response: We have corrected this typo.

10) Eq S6: on the left 'PP' should not be in capital letters.

Our response: We have modified the 'PP' on the left in Eq. S6 to be in lowercase letters.

11) Fig. S1: the label '4000' is cut and it looks like '400'.

Our response: We have corrected the label in this figure as follows:

Figure S2. Comparison of the harmonic IR spectra of the $\text{H}^+(\text{H}_2\text{O})_{21}$ cluster in the Eigen state computed at the CCD and MP2 levels, respectively, by using the aug-cc-pVDZ basis set. The MP2 result was based on the optimized low-lying energy isomer from ref. 33.

At the light of the considerations reported above, I think that the work presented is mainly of incremental type and more suitable for a specialistic journal, as it has been the case of other publications that I cited above. Finally, the authors should clean their draft from the numerous overstatements and report their results in the form of stick spectra.

Our response: We thank the reviewer for his/her fair scientific evaluations on our work. We have found the comments very constructive and helpful. In the revised manuscript, we have cleaned the draft from those overstatements and added the stick spectrum in the Supplementary Materials.

Reviewer #2 (Remarks to the Author):

The protonated water clusters have been the subject of extensive experimental and theoretical studies over past decades, due to their great importance on understanding the fundamental role these simple ions play in aqueous chemistry. This manuscript provides a complete and clear assignment of the experimental spectroscopic features of protonated water cluster $\text{H}^+(\text{H}_2\text{O})_{21}$ based on high-level electronic wavefunction theory and anharmonic vibrational perturbation theory, and contributes to fully understanding the microscopic behavior of the excess proton in the water cluster at the atomic level. I thoroughly enjoyed the insights provided by this manuscript, and strongly believe it is worthy of publication, particularly as it extends and adds to the discussion of spectral signatures of ice-like and liquid-like water molecules. Overall, it is a very impressive achievement and I do believe that this paper will be of great interest to a broad range of the Nature Communications readers.

Therefore, I recommend its publication in Nature Communications after a minor revision by considering the following comments.

1) The high-level CC calculations of the complex $\text{H}^+(\text{H}_2\text{O})_{21}$ cluster were carried out using the authors' previously developed EE-GMF fragmentation method. The authors may briefly address the accuracy and efficiency of this method in the main text, so that the readers could know the computational cost to achieve such high-level calculations. In addition, is the EE-GMF program open-source, and could anyone who is interested in this method use it for his/her own studies?

Our response: We thank the reviewer for making this comment.

We have addressed the accuracy and efficiency of EE-GMF on page 4 of the revised manuscript as follows:

“Our previously developed electrostatically embedded generalized molecular fractionation (EE-GMF) method(49), whose accuracy and efficiency have been rigorously evaluated in a series of studies, is utilized to reduce the computational scaling of the full system CC calculations. EE-GMF shows an acceleration by factors

of >40 over the conventional full system calculations, while the deviations of EE-GMF calculated energies of systems containing over 100 water molecules at diverse ab initio levels are mostly within 0.01 a.u. as compared to the full system calculations.(49, 53)”

49. J. F. Liu, L. W. Qi, J. Z. H. Zhang, X. He, J. Chem. Theory Comput. 13, 2021-2034 (2017).

53. J. F. Liu, X. He, Phys. Chem. Chem. Phys. 22, 12341-12367 (2020).

EE-GMF is an open-sourced program. The EE-GMF codes and related input data files have been uploaded to <https://github.com/jinfeng-data/H-H2O-21>. Anyone who is interested in this method could download it freely.

We have provided those information in the **Data and materials availability** section as follows,

“The EE-GMF codes, input and output files for quantum mechanical and vibrational calculations underlying the conclusions of this work are archived in <https://github.com/jinfeng-data/H-H2O-21>.”

2) *In the present work, the frustrated rotations of H₃O⁺ are all included in the resonance or combination bands of the asym/sym OH stretch, umbrella and bending of H₃O⁺, but where is the fundamental of the frustrated rotation of H₃O⁺ in the IR spectra? As the readers may be concerned about these fundamental vibrations, the authors should make a clear description of all the fundamental vibrations of H₃O⁺, as well as their couplings, in the main text. In addition, the authors may also give the assignment of the vibrational peaks in the librational region.*

Our response: The fundamental of the frustrated rotation calculated at 943 cm⁻¹ is now indicated in Figure 2. Note that there are two other components which are IR inactive (shown in Table S1). Their motions have different axis of rotation (shown in Figure S5).

We also give a detailed assignment of the bands in the librational region on page 8 of the revised manuscript as follows:

“The three predicted absorption peaks (696, 720, and 746 cm⁻¹), occurring below 750 cm⁻¹, arise from the libration of three neighboring DDA^h-type neutral water molecules around the surface-bound H₃O⁺ ion in the single hydrogen-bond acceptor configurations. The remaining bands in this region correspond to the libration of the AAD and AADD^s-type water molecules far from H₃O⁺. The strongest peak calculated at 855 cm⁻¹ agrees well with the experimental peak around 840 cm⁻¹. The detailed assignments of the librational bands are summarized in Table S2, and the vibrational motion is illustrated in Figure S6 of the Supplementary Material.”

Table S2. The assignments of the librational modes of neutral water.

Type	Exp./cm ⁻¹	VQDPT2/cm ⁻¹	Weight	Intensity
DDA ^h	*	696	0.348	91
AAD	*	707	0.397	52
AAD	*	714	0.235	14
DDA ^h	*	720	0.367	74
AAD	*	725	0.288	23
DDA ^h	*	746	0.459	91
AADD ^s	*	777	0.409	82
AAD	*	795	0.554	81
AAD	~840	855	0.248	293
AAD	*	862	0.379	63
AADD ^s	*	893	0.209	30
AAD	*	908	0.265	190
AAD	*	923	0.319	39
AAD	*	963	0.487	49

3) *The coupling between vibrational modes is essential for energy transfer. At present, the information on the vibrational coupling of the H-O-H bending mode of water is relatively lacking, even though the bending mode is an essential intermediate for the energy relaxation pathway. The novel energy relaxation pathway through bending overtone states found in this work provides very important information, hence the authors may discuss more on Fig. 4 to elucidate the vibrational energy transfer pathway of water.*

Our response: Although elucidation of the energy transfer pathway requires a time-dependent calculation which is beyond the scope, we have added the more details about the coupling terms and the mechanism of state mixing on page 10 as follows:

“The vibrational modes, the resonance diagram, and the component of vibrational wavefunctions are shown in Figure 4B. It is notable that the coupling constant between modes 100 and 127 (calculated as 12 cm^{-1}) is not particularly large compared to others; for example, the coupling constants of mode 100 with bending modes of DDA^d- and AADD^s-type water molecules in the nearest neighbor are obtained as 14 and 16 cm^{-1} , respectively. Instead, the state mixing is induced by the match of frequency, where the fundamental excitations of the two AAD-type water molecules are obtained as 1605 and 1609, whereas those of DDA^d- and AADD^s-type water molecules are higher in frequency at 1640 and 1651 cm^{-1} . The result implies a novel relaxation pathway of the O-H stretching excitation energy through bending overtone states mediated by the vibrational resonance.⁽⁶¹⁾ These peaks are observed around 3230 and 3110 cm^{-1} in the experiment.”

61. S. Imoto, S. S. Xantheas, S. Saito, *J. Phys. Chem. B* **119**, 11068-11078 (2015).

4) *In the comparison of the OH stretching features of different low-lying energy structures, there are a few unassigned signatures (denoted in black color) for isomers a2, a3 and a4 in Fig. 5C. What makes these unassigned features different from features a1? Could the authors provide more details on this?*

Our response: We thank the reviewer for making this comment. We have described these unique features for isomers a2, a3 and a4 in the Supplementary Material of the revised manuscript as follows:

“Comparison of O-H stretches of the neutral water in different minimum energy structures

For those unique features appeared in the O-H stretching region of neutral water in structure a2, a3 and a4 in Fig. 5C, we could also give clear assignments. For structure a2, the peak at 3461 cm^{-1} stems from the resonance of the symmetric and asymmetric O-H stretching vibrations of the AADD^s-type water. The relatively weak feature at 3360 cm^{-1} is a combination band of the bending vibrations of H₃O⁺ and AADDs-type water. For structure a3, both of the 3432 and 3470 cm^{-1} features are result of the resonance of the symmetric and asymmetric O-H stretching vibrations of the AADD^s-type water. The peak at 3421 cm^{-1} arises from the resonance of the asymmetric O-H stretching vibrations of the AADD^s-type water with the bonded O-H vibration of AAD-type water. The peaks around 3390 cm^{-1} are mixture of combination bands including the frustrated rotation of H₃O⁺, the rocking and bending vibrations of the DDA^h-type waters. For structure a4, the peak at 3466 cm^{-1} is also attributed to the resonance of the symmetric and asymmetric O-H stretching vibrations of the AADD^s-type water. The 3458 cm^{-1} feature is due to the resonance of the symmetric and asymmetric O-H stretching vibrations of the DDA^h-type water. The feature at 3450 cm^{-1} manifests the bonded O-H vibrations of AAD-type water. The peak at 3368 cm^{-1} is a combination of the bending vibrations of H₃O⁺ and DDA^h-type water. The weak 3341 feature is attributed to the symmetric O-H stretching vibrations of the AADDⁱ-type water.”

Reviewer #3 (Remarks to the Author):

In this work, high level calculations of the vibrational absorption spectrum of the H⁺(H₂O)₂₁ cluster are presented. In particular, an assignment of all the experimental absorption bands is proposed based on the calculations.

As far as I can tell, technically this work is of high quality. Combining highly accurate state of the art methods for both the electronic (fragment-based Coupled Cluster) and the nuclear (VQDPT2) problem, the frequencies of the experimental spectral bands are well reproduced. The agreement on relative intensities is less impressive (more comments on this would have been useful).

The manuscript is clearly written, and the band assignment is precisely described. My perplexity concerns the degree of broad interest of this work. By comparing the results presented here with those of the recent ref. 40 (Q. Yu, J. M. Bowman, J. Phys. Chem. A 124, 1167-1175 (2020)), I find essentially a general confirmation of the results presented there, with some improvements in the description of the bands at around 2200 cm⁻¹. As such, from the broad perspective of understanding these clusters, the present results look like refinements that do not substantially change the established picture. If this is not the case, then the important message of the work is lost amid the details of the band assignment description. In short, while the computational achievement is evident, I am not fully convinced the results themselves are of broad interest or of primary importance in the field. As such, I am not able to recommend publication in Nature Communications.

Our response: We thank the reviewer for making this comment. The new insights into the H⁺(H₂O)₂₁ cluster obtained in this study are: (1) The assignment of IR bands of the proton defect is revealed in a range of 1700 - 2800 cm⁻¹, (2) Four-coordinated water molecules on the surface (AADD^s) are liquid-like, whereas the one in the interior (AADDⁱ) is ice-like, (3) The water-water interaction invokes a relaxation of O-H stretching vibrational energy of water molecules mediated by bending overtone of nearby water molecules. We agree with the reviewer that the original manuscript lacked in conciseness. These points are now clearly mentioned in the abstract and the discussion section. In the result section, we organized subsections, “Band assignment”, “Two types of AADD water”, and “Intermolecular couplings of water molecules”. In addition, we have extensively revised the manuscript as described below.

The VCI spectrum reported by Yu and Bowman is now added in Figure 2. Figure 2 shows that some proton defect bands (e.g. peaks around 2200 and 2720 cm⁻¹) were not well resolved in VCI. In contrast, the present work reached a good agreement with the experiment providing the assignment of all characteristic bands in a range of 1700 – 2800 cm⁻¹. On the other hand, the assignment of the O-H stretching bands of water molecules in a range of 3100 – 3600 cm⁻¹ had an overlap with Yu and Bowman, and thus the description is now moved to the supplementary material to make the “Band assignment” subsection short and concise.

We have added the following information on page 9 of the Supplementary Material:

“Assignments of O-H stretches of the neutral water molecules of structure a1

The relatively isolated peak at 3595 cm⁻¹ with a shoulder around...”

More importantly, we find that the O-H stretching frequency of the AADD^s-type water molecules at the surface of the cluster is significantly higher than that of the interior AADDⁱ-type water molecule, which suggests two distinct structures of four-coordinated water molecules. The interior four-coordinated water forms near-perfect hydrogen bonds with its neighbors and thus is ice-like. In contrast, the hydrogen bonds formed between the surface four-coordinated water molecules and its partners are distorted, exhibiting the structural property similar to liquid water. This point is described in the subsection “Two types of AADD water”. The significance of this finding is emphasized in the discussion,

and on page 12:

“We emphasize that the revelation of the IR band assignments has a profound impact on the understanding of molecular structures in various systems. The precise assignments of the proton defect band in a hydrogen bonded network carve a path for addressing several open questions related to the nature of proton speciation in water. The site-specific analysis of the water O-H stretching region reveals distinct structures for the internal ice-like and surface liquid-like four-coordinated water molecules that are the cornerstone of understanding the local structure of water in diverse environments; for example, the water/air or solid interface, water clusters and droplets in amorphous polymer, and so on.”

Another unique feature of the present work is the inclusion of the water-water interaction in the vibrational Hamiltonian. In order to clarify such an effect on the IR spectrum, we have carried out the calculation with and without the water-water interaction, and compared the difference in newly added Figure 4A in the main text and Figure S8 of the Supplementary Material. This point is addressed in a subsection “Intermolecular couplings of water molecules”.

Additionally, on page 10:

“Intermolecular couplings of water molecules. Given an increasing interest in the relaxation of O-H stretching vibration in caged water clusters (60), let us address the effect of the water-water couplings on the calculated spectrum. The intermolecular coupling between the water molecules were taken into account in

VQDPT2 calculations by including bi-linear coupling terms in the PES. Note that the water-water coupling was excluded in the previous work (44). The VQDPT2 spectrum excluding the water-water couplings (except for DDA^h-DDA^h so as to retain the inter-molecular couplings of the H₃O⁺(H₂O)₃ moiety) is shown in Figure 4A for the O-H stretching region, and in Fig. S8 of the Supplementary Material for the lower frequency region. The two spectra with and without the water-water couplings gives major peaks in a similar position, and thus the overall appearance looks similar. Nonetheless, the presence of the water-water coupling generally makes the spectrum more broadened and widespread. For example, the O-H stretching band in a range of 3100 - 3400 cm⁻¹ exhibits noticeable differences. In the fully coupled model, the peaks calculated at 3284 and 3166 cm⁻¹ (denoted as **g** and **i**, respectively) manifest intra-molecular Fermi resonance of an AAD-type water molecule between an overtone of the bending mode (No. 100) and a fundamental of the O-H stretching mode (No. 101). Interestingly, the lower frequency component is further resonant with an overtone of the bending mode (No. 127) of a neighboring AAD-type water molecule. The vibrational modes, the resonance diagram, and the component of vibrational wavefunctions are shown in Figure 4B.”

44. Q. Yu, J. M. Bowman, *J. Phys. Chem. A* **124**, 1167-1175 (2020).

60. N. Yang, C. H. Duong, P. J. Kelleher, M. A. Johnson, *Nat. Chem.* **12**, 159-164 (2020).

Figure 4. (A) Comparison of the IR spectrum obtained by VQDPT2 with and without the harmonic coupling between water molecules in the PES. Note that the H₃O⁺(H₂O)₃ moiety has all inter-molecular couplings included in both cases. (B) The vibrational modes, the resonance diagram, and the component of vibrational wavefunctions for the intra- and inter-molecular Fermi resonance of the AAD-type water molecules.

Figure S8. Comparison of the IR spectrum obtained by VQDPT2 with and without the harmonic coupling between water molecules in the PES. Note that coupling between DDA^h are included in both cases, so as the H₃O⁺(H₂O)₃ moiety to have all inter-molecular couplings included.

An intriguing consequence is an energy relaxation pathway through bending overtone states of a neighboring AAD-type water molecule mediated by the vibrational resonance. The scheme was already illustrated in Figure 4B and described in the main text in the original manuscript, but the details are added in the revised text,

On page 10:

“It is notable that the coupling constant between modes 100 and 127 (calculated as 12 cm⁻¹) is not particularly large compared to others; for example, the coupling constants of mode 100 with bending modes of DDA^d- and AADD^s-type water molecules in the nearest neighbor are obtained as 14 and 16 cm⁻¹, respectively. Instead, the state mixing is induced by the match of frequency, where the fundamental excitations of the two AAD-type water molecules are obtained as 1605 and 1609, whereas those of DDA^d- and AADD^s-type water molecules are higher in frequency at 1640 and 1651 cm⁻¹.”

Finally, the technical novelty of the present work is the combination of EE-GMF fragmentation method and VQDPT2. This point is emphasized in the introduction of the main text,

On page 4:

“Our previously developed electrostatically embedded generalized molecular fractionation (EE-GMF) method (49), whose accuracy and efficiency have been rigorously evaluated in a series of studies, is utilized to reduce the computational scaling of the full system CC calculations. EE-GMF shows an acceleration by factors of >40 over the conventional full system calculations, while the deviations of EE-GMF calculated energies of systems containing over 100 water molecules at diverse *ab initio* levels are mostly within 0.01 a.u. as compared to the full system calculations.(49, 53) VQDPT2 has been tested to be as accurate as VCI for small molecules (51, 52), but is scalable to many-mode systems. Recently, the method has been further improved by utilizing local coordinates and applied to strongly hydrogen bonded network in biomolecules (Yagi and Sugita, submitted). In this work, VQDPT2 has been carried out in 89 dimensions using coordinates localized to each molecule of the H⁺(H₂O)₂₁ cluster. (See the Materials and Methods section in the Supplementary Material for details).”

49. J. F. Liu, L. W. Qi, J. Z. H. Zhang, X. He, *J. Chem. Theory Comput.* **13**, 2021-2034 (2017).

51. K. Yagi, S. Hirata, K. Hirao, *Phys. Chem. Chem. Phys.* **10**, 1781-1788 (2008).

52. K. Yagi, H. Otaki, *J. Chem. Phys.* **140**, 084113 (2014).

53. J. F. Liu, X. He, *Phys. Chem. Chem. Phys.* **22**, 12341-12367 (2020).

Finally, in the first paragraph of the Discussion section on page 11:

“We have developed a new protocol to compute the IR spectrum of molecular clusters based on the combination of EE-GMF and VQDPT2 for treating the electronic and vibrational problems, respectively. In the EE-GMF method, all monomers and dimers are calculated by the *ab initio* electronic structure calculations with the electrostatic embedding scheme. This scheme, in which the environmental effects are incorporated by surrounding atomic point charges, accounts for polarizability and hydrogen-bond cooperativity of the dimer, thereby making the truncation after the dimer terms far more accurate than a simple summation of bare monomer/dimer energies. VQDPT2 treats the strong interaction among quasi-degenerate states by VCI, and the weak interaction with many, non-degenerate states by the second-order perturbation theory. Unlike the regular perturbation theory, VQDPT2 is capable of describing resonance states without divergence while keeping the cost-efficiency and the scalability to many-mode systems. Furthermore, we employ vibrational coordinates localized to each molecule and represent the PES in terms of “intra”-molecular anharmonicity and “inter”-molecular harmonic coupling. The PES generated by EE-GMF is used for VQDPT2 calculations. The method is an ideal combination to compute the vibrational spectrum of molecular clusters, exploiting the locality of electronic and vibrational motions.”

These methods are described in more detail in the Materials and Methods of the Supplementary Material (page 3 – 9). We also added a note on the technical difference from Yu and Bowman in the Discussion section, on page 12:

“The present calculation is complementary with the previous VCI calculation based on *ab initio* PEF by Yu and Bowman. On one hand, the PEFs of $H^+(H_2O)_n$ ($n = 1 - 4$) and $(H_2O)_n$ ($n = 1 - 3$) were derived at the mixed high electronic structure levels of CC and MP2, but they were simply summed to construct the PEF of $H^+(H_2O)_{21}$, whereas the PES in our calculation is computed for $H^+(H_2O)_{21}$ by the fragment-based CC in an electrostatically embedding scheme at the level of CCD/aug-cc-pVDZ. On the other hand, VCI was carried for a $H_3O^+(H_2O)_3$ moiety in 15 dimension incorporating $\sim 140,000$ of VCI states and other water molecules in 3 dimension, whereas VQDPT2 was performed for $H^+(H_2O)_{21}$ in 89 dimension incorporating $\sim 1,000$ of quasi-degenerate states by VCI and hundreds of millions of non-degenerate states by perturbation. There are multiple measures on the level accuracy, and the two approaches are complementary with each other. Nonetheless, the resulting IR spectra exhibit an overall agreement, which substantiates the robustness of the theory even for such a complex system as $H^+(H_2O)_{21}$.”

Nonetheless, we note that the EE-GMF method is generally applicable to other systems without PES fitting, for instance, the ion-water clusters (J. Chem. Theory Comput., 13, 2021 (2017); J. Phys. Chem. B 122, 10202 (2018)), ionic liquid clusters (Phys. Chem. Chem. Phys., 19, 20657 (2017)), and condensed phase clusters with autoionization (J. Phys. Chem. Lett. 12, 3379 (2021)).

The agreement on relative intensities is less impressive (more comments on this would have been useful).

Reply: We have added a comment on the intensity in the final paragraph of page 13:

“Although the present calculation predicted the IR peak position in good match with the experiment, the agreement of the intensity and line-shape is less sufficient, in particular, in a range of $1700 - 2700 \text{ cm}^{-1}$. This is primarily because the calculated spectrum was constructed by simply augmenting the peak position and intensity using Lorentz functions of constant FWHM (5 cm^{-1}). The procedure is valid when the excited state has long lifetime. However, the broad, line-shape observed in the experiment indicates fast dynamics of the proton defect and the vibrational mode mixing after the excitation of O-H stretching vibration of H_3O^+ . We also found an indication of the inter-molecular energy relaxation pathway of the O-H stretching excitation of neutral water molecules via H-O-H bending overtones. With the advent of the experimental techniques (IR-IR hole burning, 2D-IR, etc.), revealing the fast dynamics of proton defect and water molecules is intriguing. Further theoretical improvement is needed to extend the framework to time-dependent quantum theory as well as to generate an accurate PES for quantum dynamics, which will be the scope of future works.”

We believe that the points discussed previously and the new information entered in the revised manuscript address all comments suggested by the reviewers. I wish to thank you for your time.

Sincerely,

Sotiris S. Xantheas, Ph.D.
Laboratory Fellow
Advanced Computing, Math. & Data Division
Pacific Northwest National Laboratory
Richland, WA

Affiliate Professor of Chemistry
UW – PNNL Distinguished Faculty Fellow
Department of Chemistry
University of Washington
Seattle, WA

REVIEWERS' COMMENTS

Reviewer#1

I thank the authors for the detailed replies. I think that the revised form is improved respect to the original one and that the authors did a concrete effort in making some issues I have arisen more clear. However, I still do not share some of their conclusions, in particular that one about the accuracy of their EE-GMF method and the related PES respect to Bowman's one.

Beyond that, I keep thinking that the overall new physical insights provided by this MS are not so striking to warrant the publication in Nature Communications, for the same reasons I already wrote in several part of my first revision. In the new version of the MS, the conclusions are unchanged, the author keep focusing only on the VQDPT2 approach without double checking their results with a MD-based method, and the novelty aspects which are now better explained are the same as in the first submission. I think that a sectoral journal is more appropriate for this type of MS.